# Can SARS-CoV-2 Virus Use Multiple Receptors to Enter Host Cells?

**DOI:** 10.3390/ijms22030992

**Published:** 2021-01-20

**Authors:** Laura Kate Gadanec, Kristen Renee McSweeney, Tawar Qaradakhi, Benazir Ali, Anthony Zulli, Vasso Apostolopoulos

**Affiliations:** Institute for Health and Sport, Victoria University, Melbourne 3030, Australia; laura.gadanec@live.vu.edu.au (L.K.G.); kristen.mcsweeney@live.vu.edu.au (K.R.M.); tawar.qaradakhi@live.vu.edu.au (T.Q.); benazir.ali@live.vu.edu.au (B.A.)

**Keywords:** angiotensin-converting enzyme 2, ACE2, COVID-19, c-lectin type receptor, glucose-regulated protein 78, SARS-CoV-2, spike protein, toll-like receptor, N-glycans, glycosylation, mannose receptor

## Abstract

The occurrence of the novel severe acute respiratory syndrome coronavirus-2 (SARS-CoV-2), responsible for coronavirus disease 2019 (COVD-19), represents a catastrophic threat to global health. Protruding from the viral surface is a densely glycosylated spike (S) protein, which engages angiotensin-converting enzyme 2 (ACE2) to mediate host cell entry. However, studies have reported viral susceptibility in intra- and extrapulmonary immune and non-immune cells lacking ACE2, suggesting that the S protein may exploit additional receptors for infection. Studies have demonstrated interactions between S protein and innate immune system, including C-lectin type receptors (CLR), toll-like receptors (TLR) and neuropilin-1 (NRP1), and the non-immune receptor glucose regulated protein 78 (GRP78). Recognition of carbohydrate moieties clustered on the surface of the S protein may drive receptor-dependent internalization, accentuate severe immunopathological inflammation, and allow for systemic spread of infection, independent of ACE2. Furthermore, targeting TLRs, CLRs, and other receptors (Ezrin and dipeptidyl peptidase-4) that do not directly engage SARS-CoV-2 S protein, but may contribute to augmented anti-viral immunity and viral clearance, may represent therapeutic targets against COVID-19.

## 1. Introduction

Betacoronavirus is one of four genera of coronaviruses (alpha, beta, gamma, delta), which are enveloped positive stranded RNA viruses, and require humans and other mammals as hosts to replicate. In particular, the natural host of betacoronaviruses are rodents and bats [1,2]. These viruses are of particular clinical importance for humans as they are known to infect humans and cause disease, including human coronavirus HKU1 and OC43, which cause the common cold, and severe acute respiratory syndrome coronavirus (SARS-CoV), and SARS-CoV-2, which is responsible for coronavirus disease 2019 (COVID-19) [1,3]. Middle East respiratory syndrome-related coronavirus (MERS-CoV) is also a betacoronavirus that first affected humans in 2012. The World Health Organization declared SARS-CoV-2 as an infectious outbreak in March 2020 [3], and it has since spread to over two hundred countries [4]. To date, COVID-19 is responsible for over 42 million infections and over 1.14 million deaths worldwide (as of 23 October 2020) [5], and has had significant detrimental effects on the global economy [6]. Like most general infections, SARS-CoV-2 induces mild symptoms in the general population [7]. However, a subset of patients, specifically those with pre-existing comorbidities, including hypertension, diabetes, and cardiovascular disease (CVD), are at a greater risk of developing severe COVID-19 and/or death [8,9]. The renin-angiotensin system has been implicated in the pathogenesis of hypertension, diabetes, and CVD [10,11]. Angiotensin II (AngII), a predominate metabolite of the renin-angiotensin system, is a potent vasoconstrictor mediating hypertension and is associated with vascular complications in patients with diabetes and CVD [12,13]. The production of AngII is counterbalanced by angiotensin-converting enzyme 2 (ACE2), which is utilized by SARS-CoV-2 to enter and infect alveolar epithelial cells [14]. As such, it has been suggested that the activation of the renin-angiotensin system could pre-dispose the patients with comorbidities to severe COVID-19 [15,16]. There was a theoretical endorsement that the increase of ACE2 by the indirect effects of AngII receptor blockers (ARB) and ACE inhibitors taken by patients with CVD and related comorbidities, could enhance docking sites for SARS-CoV-2, leading to severe COVID-19 [17,18]. Recently, a randomized clinical trial was conducted to determine whether ARBs and ACE inhibitors taken by patients with CVD, contributed to the progression of severe COVID-19 [19]. This 30-day trial demonstrated that suspending (n = 334) or continuing (n = 325) these therapies had no effect on the mortality rate of COVID-19 in this population [16,19], suggesting that the indirect increase of ACE2 does not increase intracellular viral load, and is not attributed to severe COVID-19. Yet, patients with pre-existing CVD have a higher mortality rate and are five-times more likely to experience severe COVID-19 symptoms when compared to other COVID-19 populations [20]. Thus, indicating that SARS-CoV-2 may employ additional receptors to gain entry into and infect cells, independent of ACE2 expression.

SARS-CoV-2 is a lipid-enveloped positive-sense RNA virus [1,21], which uses a heavily glycosylated spike (S) protein to facilitate its attachment, membrane fusion, and entry into host cells [1,2,22]. The S protein is a class I fusion polypeptide chain consisting of 1273 amino acids, which associates as a homotrimer [1,23]. The individual monomers are composed of two subunits, subunit 1 (S1) and subunit 2 (S2), which contain three distinctive topological domains: (i) Head, (ii) central stalk, and (iii) cytoplasmic tail [1]. The S1 contains an N-terminal domain and the receptor-binding domain, which houses the receptor-binding motif [1,24]. The receptor-binding motif has been shown to interact with ACE2, permitting entry and infection of host cells [14,25,26]. The S2 contains a central helix, connecting domain and a fusion protein, which facilitates union between viral and host cell membranes [1]. Located between S1 and S2 is a unique furin cleavage site, postulated to be cleaved by transmembrane serine protease 2 [1,23,26] for S protein priming, which is necessary for infection [1,27]. An additional proteolytic cleavage site located in the S2 is utilized for release of the fusion peptide, allowing for host cell penetration and fusion [1]. As shown by previous literature, protein glycosylation is crucial for viral infection [28,29]. SARS-CoV-2 S protein is extensively glycosylated [1,2,22,23], sharing approximately 67% of sequence homology [23] and 18 N-glycosylated sites with SARS-CoV [1]. Mapping of SARS-CoV-2 S protein has led to the discovery of 2 predicted O-glycosylation sites [22] and 22 predicted N-glycosylation predicted sites per promoter [23,30]. N-glycan moieties, present on the S protein, are responsible for receptor binding domain conformational changes necessary for association with ACE2 [14,25,26]. However, studies have reported viral susceptibility in intra- and extrapulmonary immune and non-immune cells with ACE2 deficiency [31,32,33] or absence [32], suggesting that the S protein exploits additional receptors to gain entry into host cells leading to systemic infection [2,34,35].

Burgeoning evidence is emerging from in vitro and in silico models, demonstrating interactions between S protein and immune receptors, including Neuropilin-1 (NRP1), C-lectin type receptors (CLR) (mannose receptor (MR); dendritic cell-specific intracellular adhesion molecule-3-grabbing non-integrin (DC-SIGN); homologue dendritic cell-specific intercellular adhesion molecule-3-grabbing nonintegrin related (L-SIGN); and macrophage galactose-type lectin (MGL)) [34,35] and toll-like receptors (TLRs) (TLR1; TLR4; and TLR6) [2], and the non-immune receptor glucose regulated protein 78 (GRP78) [36]. Furthermore, Ezrin [37], and dipeptidyl peptidase-4 (DPP4) [38] have been postulated to be targets against SARS-CoV-2, but have yet to be confirmed in in vitro and in vivo models. Thus, these receptors and proteins may represent alternative routes for viral infection by facilitating receptor-dependent internalization of the S protein (Figure 1). We postulate that recognition of mannosylated N-glycan and O-glycan moieties present on the S protein [1,22] by GRP78 [36] and specific CLRs [34,35] and TLRs [2,39,40] may facilitate association and internalization of S protein, and may account for accentuate severe immunopathological inflammation (resulting in cytokine release syndrome). Additional CLRs (blood dendritic cell antigen-2 (BDCA-2); C-type lectin-like receptor 2 (CLEC2); dendritic cell-associated C-type lectin-1 (Dectin-1) and -2 (Dectin-2); dendritic cell immunoreceptor (DCIR); dendritic cell natural killer lectin group receptor-1 (DNGR1); lectin-like oxidized low-density lipoprotein receptor-1 (LOX-1); and liver and lymph node sinusoidal endothelial cell C-type lectin (LSECtin)) and TLRs (TLR3; TLR5; TLR7; and TLR8), which do not directly interact with the S protein, may have roles in other aspects of COVID-19 infection. This article provides an extensive review of current literature pertaining to the CLRs, TLRs, and other proteins that may be involved in COVID-19 and should therefore be considered as potential therapeutic targets.

## 2. Non-Immune Receptors Involved in Coronavirus Disease 2019

### 2.1. Angiotensin-Converting Enzyme 2

Angiotensin-converting enzyme converts the decapeptide angiotensin I into the octapeptide AngII, which stimulates the AngII type 1 receptor to suppress vasoconstriction. However, it also results in increased fibrosis, inflammation, thrombosis, and pulmonary damage. To balance the production and actions of AngII, ACE2 converts AngII, thereby decreasing its levels, to produce the heptapeptide angiotensin (1-7) (Ang1-7). Ang1-7 then stimulates the Mas receptor, which has been shown to be protective against lung injury [41,42]. The imbalance and loss of ACE2 and Ang1-7 leads to the increase of AngII, promoting acute lung injury [41], and is generally noted in patients with CVD. Therefore, this may account for patients with CVD having more severe COVID-19 symptoms [41]. It has been shown that SARS-CoV-2, like with SARS-CoV, binds to ACE2 for host cell entry [25,26,43] (Figure 2A). The receptor binding domain located on the outer structure of SARS-CoV-2 S protein S1 contains a core receptor-binding motif [24]. The receptor-binding motif binds to the N-terminal extracellular catalytic ectodomain, also known as the peptidase domain of ACE2, resulting in a SARS-CoV-2/ACE2 complex [25]. Upon transmembrane serine protease 2 activation by SARS-CoV-2 for S priming, the SARS-CoV-2/ACE2 complex undergoes endocytosis and forms an endosome [26]. The endosome is then acidified, resulting in the release of the encapsulated single stranded RNA of the virus into the cytoplasm for replication and translation [26]. At the same time of the abovementioned process, critically ill patients infected with SARS-CoV-2 have shown significantly elevated plasma concentrations of AngII [44]. SARS-CoV-2 stimulates nuclear factor kappa-light-chain-enhancer of activated B cells (NF-kB) driven activation, which results in the release of tumor necrosis factor alpha (TNF-alpha) and interleukin (IL)-6 in engineered human lung cell lines expressing ACE2 [45] and IL-1beta in peripheral human monocytes infected with SARS-CoV-2 [46]. Together, these findings suggest the activation of AngII/AngII type 1 receptor trigger pro-inflammatory cytokines and inflammation and possibly the inactivation of Ang1-7/Mas R pathway may be the downstream mechanism of SARS-CoV-2 upon ACE2 downregulation in patients with COVID-19.

### 2.2. Glucose-Regulated Protein 78

Glucose regulated protein 78 (GRP78; also referred to as heat shock protein A5 or binding immunoglobulin protein) is an essential endoplasmic reticulum (ER) chaperone protein, involved in maintenance and protein surveillance by controlling the unfolded protein response (cellular stress response initiated by accumulation of unfolded or incorrectly folded proteins) [36,50]. Under normal conditions, GRP78 is localized to the lumen of the ER, bound to inactivating enzymes, including activating transcription factor 6, inositol-requiring enzyme 1, and protein kinase R-like endoplasmic reticulum kinase, which are responsible for inhibiting protein synthesis, enhancing protein folding, and initiating cell death [36]. Accumulation of unfolded or misfolded proteins results in GRP78 releasing from its receptors and translocating to the plasma membrane [36]. Once translocated, GRP78 has the ability to recognize and mediate entry of viruses via the substrate-binding domain [49]. Thus, GRP78 has been investigated as a potential gateway for viral entry in COVID-19 by binding to motifs on the S protein [36,47,48].

Patients infected with SARS-CoV-2 virus have increased gene expression and serum concentrations of GRP78 [47]. It is suggested that damaged airway epithelial cells release GRP78 in response to severe pulmonary trauma and injury during SARS-CoV-2 infection [47,48], which may account for increased inflammation in COVID-19 patients (Figure 2B). In addition, GRP78 has been identified as a DAMP for specific TLRs (TLR2 [51] and TLR3 [52]) and may account for promotion of augmented inflammation in COVID-19 patients [47,48]. As previous literature has demonstrated the ability of GRP78 to interact and mediate viral entry into cells, a predictive in silico study was performed to determine its affinity for the SARS-CoV-2 S protein [36]. Four cyclic regions and their corresponding residues (region I: C336-C361, 26 residues; region II: C379-C432, 54 residues; region III: C391-C525, 135 residues; and region IV: C480-C488, 9 residues), present on the outer surface of the S protein, were selected for molecular docking assessment [36]. These regions were selected, as they have previously been targets of antibody neutralising therapies against SARS-CoV and MERS [36]. Data showed a preferred binding between regions III and IV of the S protein and the substrate binding domain-beta of GRP78 [36]. Furthermore, region IV was determined as the major driving force for GRP78 interaction, with the predicted binding affinity of −9.8 kcal/mol [36]. Therefore, as GRP78 is able to bind to the S protein of SARS-CoV-2 and has previously been shown to initiate internalization of viral pathogens, GRP78 represents a potential therapeutic target to be used in COVID-19 treatment.

### 2.3. Ezrin

Ezrin (also known as cytovillin or villin-2) is a protein that belongs to the ezrin-radixin moesin family, and is encoded by the *EZR* gene [53]. Ezrin regulates inflammation, with genetic deletion of ezrin in B cells linked to heightened expression of key anti-inflammatory markers [54]. The role that Ezrin plays during viral infection and transmission has been studied in human immunodeficiency virus-1 (HIV-1) [55]. As such, Ezrin enhances viral infectivity, through inhibition of unnecessary membrane fusion [55]. Contrary to this, in relation to SARS, previous studies noted that Ezrin interacts with the SARS-CoV spike protein through binding to the carboxy-terminus using its FERM domain [37], resulting in reduced viral entry [56]. This highlights a potential therapeutic option to prevent SARS-CoV-2 infection. In addition to inhibiting the key receptors involved in COVID-19, such as ACE2 and the newly suggested TLRs, an Ezrin agonist or molecule that increases Ezrin functionality could be a strategic approach to inhibiting SARS-CoV-2 viral entry. This hypothesis was investigated using Ezrin peptides, which have previously demonstrated effectiveness in treating a variety of viral infections, initiated by HIV-1, hepatitis C virus, human papillomavirus, herpes simplex I and II, and the causative viral agents in acute viral respiratory infection [37]. Specifically, it is particularly beneficial in inhibition of inflammation in viral pneumonia [37], a key pathophysiological complication observed in COVID-19. This could be a promising avenue to prevent acute lung injury and additional lung pathologies observed in patients with COVID-19. These data present that both activating or inhibiting Ezrin are both beneficial against inflammatory diseases. Hence, further research needs to be undertaken to delineate its role against SARS-CoV-2 viral infection.

## 3. Toll-Like Receptors Participating in Coronavirus Disease 2019 Pathogenesis and Progression

### 3.1. Introduction to Toll-Like Receptors

The innate immune system facilitates the first-line protective mechanisms against invading pathogens [57,58]. Integral to innate immunity is a superfamily of germline-encoded proteins, named PRRs [59,60]; of which, TLRs are integral proteins that provide host surveillance by detecting foreign- and self-molecular signatures [59,60]. TLRs are transmembrane type I glycoproteins, containing three structural components: (i) An N-terminal intracellular toll-interleukin 1 receptor domain, required for signal transduction, (ii) a central transmembrane domain, and (iii) an extracellular C-terminal rich in leucine repeats, which provides diversity between individual TLRs [61]. TLRs are able to identify a repertoire of pathogen-associated molecular patterns (PAMP) that respond by inducing a robust inflammatory response in order to neutralize, and eliminate invading pathogens [59,60]. In addition, TLRs respond to danger-associated molecular patterns (DAMP), which are secreted by damaged, stressed, or necrotic cells, independent of infection [62,63]. The end product of inflammation, produced through the myeloid differentiation factor-88 (MyD88)-dependent pathway (TLR1, 2, 4-10) [64] or the toll/IL-1-domain-containing adapter-inducing interferon-beta (TRIF)-dependent pathway (TLR3 and 4) [65], is ubiquitous among TLRs, independent of the origin of the activating ligand. The expression of TLRs have been reported to be present throughout the human respiratory system [57], displaying heterogeneity in specific cell populations (Figure 3). TLRs residing on the cell surface have been suggested as potential therapeutic targets in COVID-19, as a molecular docking studies have demonstrated direct binding between S protein and TLR1, 4 and 6 [2]. Furthermore, TLRs (TLR3; TLR7; and TLR8) located on the membranes of intracellular organelles (endosomes; lysosomes; endolysozomes), which are responsible for recognition of pathogenic nucleic acids [59,66], may aid in viral clearance of SARS-CoV-2. A tailored pharmaceutical regime or vaccination containing specific TLR agonists and antagonists may provide a strategic approach to dampening the exacerbated immune response, preventing systemic spread of infection and enhancing viral immunity and clearance in COVID-19 patients. We further discuss the role that specific TLRs have in SARS-CoV-2 infection below.

### 3.2. TLR1/2/6 as Potential Therapeutic Targets and Alternative Viral Entry Points for SARS-CoV-2

TLR1 and 6 cooperate with TLR2 to form functional heterodimers for receptor activation [59,86], and predominantly identify invading Gram-positive bacteria (tri- [87] and diacylated lipopeptides [88]), mycobacteria [89], and fungi [90]. Additionally, TLR1/2 [91,92] and 6/2 [92,93] heterodimers have been shown to contribute to augmented pro-inflammatory responses during viral infection through recognition of specific viral glycoproteins. Thus, indicating a potential, yet limited role in antiviral immunity. The immunopathological roles that TLR1 and 6 have during COVID-19 infection remain elusive [92]. However, elevated levels of TLR1/2/6 DAMPs, including beta-defensin-3 [94] (identified by TLR1/2 [95]) and high-mobility group box-1 (HMGB1) [96] (identified by TLR1/2/6 [97]), have been reported in peripheral blood mononuclear cells and serum collected from COVID-19 patients, respectively. Direct association between DAMPs and their corresponding TLRs are able to activate TLR-mediated inflammatory response, identical to those produced through PAMP recognition [62,63]. Thus, TLR1/2/6 activation and subsequent signal transduction may be in part responsible for clinical immunopathological manifestations experienced by patients infected with COVID-19 (Figure 4).

An in silico study showed the TLR-binding efficacy of S protein by direct binding of SARS-CoV-2 S protein to TLR1 and TLR6 [2]. Hydrogen bonding and hydrophobic interactions were evident between the interface of TLRs and the S1 of S protein, displaying binding energy values of −57.3 (TLR1) and −68.4 (TLR6) [2]. The oligomannose-type glycans present on the outer surface of the S protein may facilitate association between TLR1 and TLR6, as previous literature has demonstrated binding between TLR1/2 and TLR6/2 heterodimers with phosphatidylinositol mannosides (biosynthetic precursors of lipoarabinomannan, a component of mycobacteria cell wall) [98] and mannose-capped lipoarabinomannan [99]. SARS-CoV-2 may be a PAMP of TLR1/2 and 6/2 heterodimers, as interactions between TLR1/6 and the S protein may play a role in immunopathologies resulting from unregulated TLR activation (Figure 4). Supporting evidence has shown that elevated levels of TLR4 downstream signaling molecules have been observed in patients with COVID-19 [100]. As all TLRs (excluding TLR3) use the MyD88-depenednt pathway for signal transduction and initiation of pro-inflammatory cytokine and chemokine release [64], these results may indicate an increase in the TLR/MyD88-dependent pathway, rather than increased activation of a single TLR. Thus, it could be hypothesized that the upregulation in TLR downstream signaling molecules could be due to increased TLR/MyD88/NF-kB pathway in COVID-19 patients, of which TLR1 and 6 may participate through activation caused by beta-defensin-3, HMGB1 and S protein oligomannose-type glycans. Furthermore, TLR1/2/6 play a role in mediating viral entry upon recognition of viral fragments, though CD36- [101] (bacterial infection) or clathrin-dependent endocytosis (Dengue virus infection) [102]. Thus, it can be postulated that TLR1/2 and TLR6/2 heterodimers could be trafficked form the cell membrane into the intracellular environment, providing a gateway in which S protein can gain entry into cells. Taken together, TLR1/2 and TLR2/6 heterodimers may represent a therapeutic target of inhibitory pharmaceuticals to prevent excessive inflammation and viral entry. The recent discovery of MMG11 (a TLR2 inhibitor, which shows preference for the TLR1/2 heterodimer [103,104]) and CuCpt22 (a TLR1/2 heterodimer inhibitor in mice [103] and a TLR1/2/6 inhibitor in humans [103,105]) may represent potential COVID-19 therapeutics. Pre-treatment of MMG11 (5 μg/mL) followed by infection with *Mycobacterium avium* subspecies *paratuberculosis* in human macrophages, resulted in significantly reduced concentration of pro-inflammatory cytokines IL-8 and TNFα [104]. Similar anti-inflammatory abilities have been observed in human primary bronchial epithelial cells pre-treated with CuCpt22 (50 μM), 30 min before challenge with *Streptococcus pneumonia* strain D39 [105]. Pre-treatment with CuCpt22 was able to reduce gene expression of IL-6 and granulocyte-macrophage colony-stimulating factor and lowered expression of nuclear factor kappaB inhibitor-ζ [105] (an essential regulator of the TLR response, with increased expression being associated with pulmonary pathologies caused by exacerbated and unregulated inflammation [106,107]). However, in vitro and in vivo studies investigating the ability of MMG11 and CuCpt22 to be administered as either a prophylactic or treatment during active SARS-CoV-2 infection are required to determine: (a) The optimal time point for the greatest beneficial effect; (b) the ability to prevent viral entry into cells; and (c) the extent to which they can dampen the inflammatory response.

### 3.3. TLR3 as a Potential Therapeutic Target in SARS-CoV-2 Infection

TLR3 is responsible for antiviral immunity by recognising and interacting with viral PAMPs, including double-stranded ribonucleic acid (dsRNA) (produced during viral replication by positive sense-strand RNA and DNA viruses) [108,109], small interfering RNA [110], and incomplete stem structures present in single-stranded RNA [111]. Furthermore, liberated cellular debris, including cytoplasmic nucleotides (messenger RNA [112] and dsRNA [113]) and GRP78 [52], from host cells are activating DAMPs of TLR3. TLR3 is unique, as it is the only TLR that exclusively interacts with TRIF, leading to the activation of NF-κβ and interferon (IFN)-regulatory factor-3 and -7 [87,109]. This results in the release of pro-inflammatory molecules, including those involved in COVID19 immunopathological manifestations [8] (IL-1beta; IL-6; IL-8 and TNF-alpha [72]). Direct engagement between TLR3 and the S protein of SARS-CoV-2 has yet to be established. However, SARS-CoV-2 is a positive RNA virus [1,21], and its products released during viral replication may be identified by TLR3. Therefore, TLR3 may represent a therapeutic target, which upon activation may contribute to increased antiviral immune responses, reduce viral burden and facilitate elimination of SARS-CoV-2. Previous studies involving animal models infected with mouse-adapted SARS-CoV (MA15) [114,115,116,117] showed a protective role of the TLR3 pathway. In addition, improved viral clearance and enhanced anti-viral immunity using polyinosoinic-polycytidylic acid (Poly(I:C)) (synthetic double dsRNA analogue, which specifically activates TLR3) in aged mice infected with MA15 [114]. Intranasal pretreatment with Poly(I:C) protected mice from lethal infection (decreasing mortality rate from 80–100% to 0–10%), significantly reduced viral load within twenty-four hours, and resulted in complete viral clearance by day 10 post-infection [114]. Furthermore, 12-month-old mice pretreated with Poly(I:C) had significantly improved pathological changes, consistent with acute lung injury recovery [114]. Poly(I:C) pretreatment reduced the presence of airway debris, epithelial cell necrosis, pulmonary oedema, and infiltrating eosinophils and neutrophils [114]. Recovery from pulmonary pathologies experienced by pretreated mice may have resulted from Poly(I:C) priming and augmenting antiviral immunity through the TLR3 pathway [114]. This was demonstrated by Poly(I:C) promoting early infiltration of respiratory dendritic cells (rDC), alveolar macrophages, perivascular, and –bronchial cells, enhancing virus-specific T cell responses and facilitating premature release of pro-inflammatory molecules (IFN-beta; IFN-gamma; IL-1beta; melanoma differentiation-associated protein 5; retinoic acid-inducible gene I; signal transducer and activator of transcript-1 (STAT-1); and TNF), which resulted in the activation INF-stimulated genes (protein kinase R and 2′-5′-oligoadenylate synthetase 1) [114]. In another study, it was noted that there was enhanced anti-viral immunity in mice pretreated with ampligen (a synthetic, Poly(I:C) analogue, which activates TLR3 and induces IFN production) [115]. Mice injected with ampligen, four hours prior to MA15 infection, had undetectable pulmonary viral loads within seventy-two hours post-infection [115]. Finally, a study involving different dosing regimes of hiltonol (a synthetic dsRNA complex, which stimulates TLR3), administered to mice at various times before and after exposure to MA15, supports previous findings of the protective role of TLR3 during viral infection [116]. All doses of hiltonol, administered intranasally to different mice groups at various time points, protected mice against lethal MA15 infection and weight loss experienced during viral challenge [116]. In addition, injection of hiltonol at least eight hours following infection resulted in significantly increased survival, when compared to mice who received the treatment 16 to 72 h post infection [116]. The protective role that TLR3 has during MA15 infection is further supported in a study using TLR3 knockout (TLR3^−^/^−^) mice [117]. TLR3^−^/^−^ mice, intranasally infected with MA15, had a significantly increased viral burden when compared to controls [117]. Furthermore, TLR3^−^/^−^ mice had significantly impaired pulmonary function, demonstrated by unresolving airway resistance and obstruction parameters, which may have resulted from increased alveolar exudates observed during histological assessment [117]. Similar results have been reported in human alveolar epithelial cells, suggesting translatability between beneficial TLR3 agonists in animal models to human subjects [114]. Alveolar epithelial cells, derived from patients infected with SARS-CoV, reported reduced viral burden by 100-fold, 48 h post-treatment with Poly(I:C) [114]. Taken together, targeting TLR3 may prime the immune system, resulting in rapid antiviral responses and enhanced viral clearance during viral challenge.

Recently, type I IFN susceptibility has been exhibited by SARS-CoV-2, resulting in reduced viral loads and replication when given as a pretreatment [118,119]. Serum profiling of COVID-19 patients noted reduced levels of type I and III IFNs, suggesting that SARS-CoV-2 can reduce IFN levels to evade antiviral innate immune responses [120]. As TLR3 is a major producer of type I and III IFNs [121], activating TLR3 and restoring IFN concentrations during SARS-CoV-2 could be a viable treatment for COVID-19 patients. Human alveolar epithelial cells pretreated with type I IFN-alpha (1000 U/mL) two hours prior to SARS-CoV-2 infection, resulted in a two- and four-fold reduction in viral burden 24 and 48 h post-infection [118]. It is suggested that this reduction in viral replication is due to increased STAT-1 production and phosphorylation arresting viral replication [118]. However, type I IFN-alpha may only be beneficial as a pretreatment, as its ability to reduce viral load is limited when administered after established SARS-CoV-2 infection [118]. Vero cells administration IFN-alpha, two hours after SARS-CoV-2 infection, displayed a two-fold decrease in viral load [118]. However at 48 h viral titers exhibited similar levels to untreated groups [118]. Comparable results were reported in a study investigating pretreatment with IFN-alpha and IFN-beta in SARS-CoV-2 infection [119]. Sixteen hours before infection, vero cells were pretreated with various concentrations (50–1000 IU/mL) of recombinant human IFN-alpha and IFN-beta, and were then incubated with SARS-CoV-2 [119]. IFN-alpha was able to potently inhibit SARS-CoV-2 infection, as virus titers were undetectable at all concentrations, excluding cells administered the lowest dose [119]. However, this dose was still able to significantly reduce the viral burden four-fold [119]. Pretreatment with IFN-beta resulted in a more potent inhibition of SARS-CoV-2 infection, as viral titers were undetectable at all concentrations [119]. Additional experiments were conducted to determine the lowest concentration, which still provided significant antiviral results. [119]. Both IFN-alpha and IFN-beta displayed dose-dependent inhibition against SARS-CoV-2 infection, significantly reducing viral titers at 5 IU/mL and 50 IU/mL, respectively [119]. Hence, we postulate that repurposing TLR3 agonists (e.g., Poly(I:C); ampligen; hiltonol) as prophylactic agents, prescribed to individuals vulnerable to SARS-CoV-2 before established infection, is an appealing approach. This hypothesis is supported by literature, which demonstrates priming of anti-viral innate immunity and premature release of type I IFNs (IFN-alpha and IFN-beta) through the TRL3/TRIF pathway may accentuate antiviral immunity against SARS-CoV-2, protect pulmonary tissue from damage, and accelerate viral clearance [114,115,116,118,119].

### 3.4. TLR4 as a Potential Therapeutic Target and Alternative Viral Entry Point in SARS-CoV-2 Infection

TLR4 is primarily responsible for gram-negative bacterial immunity, through physiological recognition of lipopolysaccharides [122,123]. However, engagement and subsequent activation of TLR4 by viral fusion proteins and glycoproteins, including viruses that target the respiratory system [124,125], have been noted. Additionally, TLR4 is able to respond to a plethora of host-derived DAMPs [126], which have been shown to drive exacerbated and unregulated inflammation in chronic inflammatory and autoimmune diseases [127]. Uncontrolled TLR4-mediated inflammation has been suggested to contribute to immunopathological consequences in COVID-19 patients [100]. Peripheral blood mononuclear cells harvested from patients infected with SARS-CoV-2 have increased expression of TLR4 and its downstream signaling adapter molecules (CD14; MyD88; tumor necrosis factor-associated factor 6; IL-1 receptor-associated kinase 1; toll/IL-1 receptor domain-containing adaptor protein; TIR domain-containing adaptor molecule 1; and NF-kB) [100]. In addition, elevated levels of circulating TLR4 DAMPs in COVID-19 patients (beta-defensin-3 [94]; fibrinogen [128,129]; heat-shock protein 70 [130]; HMGB1 [96]; syndecan [131]; S100A8/9 [96,100,132]; and surfactant A and D [133]) may be responsible for the feed-forward loop of the persistent inflammation, resulting in cytokine storm. This is supported by reports of patients with COVID-19 displaying increased levels of cytokines and chemokines [9], which have been shown to be released by TLR4 activation in pulmonary pathologies (IL-1beta, IL-2, IL-6, IL-8, IL-9, TNF-alpha, granulocyte colony-stimulating factor; granulocyte-macrophage colony-stimulating factor; macrophage inflammatory protein-1 alpha, and beta [72,134]). Therefore, TLR4 activation may be responsible for immunopathological manifestations experienced by COVID-19 due to increased TLR4 expression and circulating DAMPs (Figure 4).

In silico studies investigating the TLR-bindng efficacy of S protein, have demonstrated cell surface TLR-S protein engagement, consisting of hydrogen bonding and hydrophobic interactions [2]. Of which, TLR4 displayed the highest affinity for S1 of the S protein, generating a binding energy value of -120.2 [2]. This may be due to TLR4 interacting with oligomannose- and complex-type glycan structures located on the surface of the S protein. Previous literature has determined the ability of TLR4 to recognize and be activated by lipoglycan structures [122,123,135,136], mannosylated polypeptides [137,138], and viral glycoproteins [125,139,140]. Recognition of S protein by TLR4 may also initiate receptor-dependent internalization, accounting for SARS-CoV-2 infection in patients and cells that lacking or deficient in ACE2 expression. TLR4 endocytosis has yet to be investigated in viral infection. However, it has been reported during bacterial infection, allowing microbial products to enter into cells [141,142,143]. We propose that SARS-CoV-2 S protein is a PAMP internalized by TLR4, through identification and interaction with surface glycan and mannose carbohydrate motifs on the S1. This may be responsible for increased TLR4 expression and inflammation in COVID-19 patients. Furthermore, due to the ability of TLR4 to internalize pathogens, it could represent a new viral entry point exploited by SARS-CoV-2 independent of ACE2 expression. Thus, inhibition of TLR4 should be investigated as a potential therapeutic in COVID-19 infection.

### 3.5. TLR 5 as a Potential Vaccine Target in Coronavirus 2019

TLR activating ligands have previously been used, as immunogenic adjuvants during vaccine development, to enhance vaccine efficacy and facilitate production of a tailored and robust immune responses [145,146]. Flagellin (a microtubule-based structural whip-like filament that enables locomotion in motile Gram-negative and positive bacteria [147]) is a potent immunomodulatory agent [148,149], which has been utilized as an adjuvant component in vaccine formulation due to its ability to influence pathogenic virulence [150,151]. Flagellin exclusively interacts with TLR5, and results in subsequent NF-kB driven inflammation, through recruitment of MyD88 [59,60]. Targeting TLR5 with flagellin has been investigated for development of vaccinations against viral pathogens [152,153,154,155]. The ability of TLR5 to interact with SARS-CoV-2 has yet to be fully elucidated. An in silico study, determining the direct association between surface TLRs and S protein of SARS-CoV-2, reported a positive energy for TLR5, indicating a possible TLR5-S protein interaction [2]. However, future studies are required to further investigate the ability of TLR5 to engage with SARS-CoV-2 protein [2]. Additionally, TLR5 has been suggested as a possible SARS-Cov-2 vaccination target [156,157], as early TLR5 activation may improve anti-viral immunity (thorough production of type I IFNs [158]) and maturation of rDC [159]. Recent literature has demonstrated an epitope-based peptide vaccine component against SARS-CoV2, which successfully docked with TLR5 [156]. An additional study developed a SARS-CoV-2 subunit recombinant vaccine against S1, which resulted in TLR5 activation [157]. Altering anti-viral immunity through immunomodulation of TLR5 may represent a strategic vaccination approach, by priming the innate immune system to produce augmented anti-viral mechanisms (type I IFN release) against viral replication after entry into host cell. However, TLR5 activation may participate in severe COVID-19 complications, as these patients have increased serum levels of HMGB [96] (a DAMP of TLR5 [160]). Thus, overstimulation of TLR5 may result in increased pro-inflammatory release and development of cytokine storm. Further research needs to be conducted to determine the role that TLR5 plays during SARS-CoV-2 to determine if its activation is detrimental.

### 3.6. TLR7/8 as Potential Therapeutic Targets for SARS-CoV-2 Infection

As with TLR3, TLR7, and TLR8 are PRRs located on intracellular organelles [66,161,162], which provide anti-viral immunity through recognition of viral single-stranded RNA (ssRNA) and elicitation of subsequent pro-inflammatory mechanisms [163,164]. TLR7 and TLR8 are often referred to collectively, as they share a high degree of sequence homology [165] and similar functionality [166]. However, discrepancies between the two receptors have been reported [167]. The ability of TLR7/8 to reduce replication of viruses has been demonstrated in HIV-1 [1], influenza [2], and MERS-CoV [168], as upon entry into the cell viral ssRNA binds to TLR7/8 promoting activation and antiviral immunity. Activation of TLR7 [163,169] and TLR8 [170] induces the recruitment of the adaptor molecule MyD88, resulting in the release of pro-inflammatory cytokines and chemokines [72], and type I (IFN-alpha and IFN-beta) and III IFNs (IFN-lambda) [171], which have been shown to aid in viral clearance and reduced replication. It remains unknown if TLR7/8 can directly interact with SARS-CoV-2 S protein, upon entry into host cells. However, they have been suggested as possible SARS-CoV-2 therapeutic targets due their anti-viral immunity and ability to sense ssRNA. Recognition of viral genomic ssRNA from positive-sense RNA viruses has been shown to be recognized by endosomal TLR7/8 [172,173,174]. A bioinformatic analysis, investigating genomic ssRNA fragments of SARS-CoV-2, reported a larger number of fragments (greater than that shown in SARS-CoV) that could be identified by TLR7/8 [175]. These results suggest that rapid release of type I IFNs by TLR7/8 could influence pathogenicity of SARS-CoV-2 by altering: DC growth, maturation and apoptosis, cytotoxicity of natural killer cells, and virus-specific cytotoxic responses produced by T lymphocytes [175]. Additionally, during adenovirus type 5 infection the TLR7/MyD88 pathway was responsible for subsequent signal transduction by lung epithelial cells, necessary for IFN production [168]. Thus, recognition of SARS-CoV-2 ssRNA by TLR7/8 may result in antiviral immunity through increased production of cytokines and IFNs. The protective and anti-viral role that TLR7 plays in SARS-CoV-2 infection is further supported by severe COVID-19 outcomes experienced in young men with X-chromosomal TLR7 genetic anomalies [176]. Inheritance of a four-nucleotide deletion (c.2129_2132del; p.[Gln710Argfs*18]) or a missense variant (c.2383G>T; p.[Val795Phe]) resulted in reduced mRNA expression of TLR7, deleterious effects on the TLR7 structure, and functionality and defective production of type I and II IFNs [176]. However, due to the simultaneous release of pro-inflammatory cytokines and chemokines, activation of TLR7/8 during SARS-CoV-2 may provoke an augmented inflammatory response, which could result in severe and potentially lethal immunopathological consequences experienced by COVID-19 patients [175]. Patients with COVID-19 have shown increased circulating levels of pro-inflammatory cytokines and chemokines, which are produced through the TLR7/8 pathways [72]. This may be due to TLR7/8 recognizing antiphospholipid antibodies (aPL) (a TLR7/8 activating DAMP [177,178,179]), which have been shown to be upregulated in COVID-19 patients [180,181]. aPLs are a family of autoantibodies that associate with negatively charged phospholipids, resulting in the disruption of self-tolerance and launch of autoimmune responses targeting host phospholipids [177,178,179]. A study investigating the presence of aPLs in severe and critical COVID-19 patients (admitted to the intensive care unit) determined that patients infected with SARS-CoV-2 had increased concentrations of circulating aPLs, when compared to healthy individuals [180]. Of the 21 COVID-19 patients recruited, at least 12 patients had increased circulating levels of at least one aPL (antiannexin V IgM: 19%; anticardiolipin IgM: 14%; antiphosphatidylserine IgM: 14% anticardiolipin IgG: 10%; and antiphosphatidylserine IgG: 10%) [180]. Thus, the exacerbated immune response resulting in cytokine storm may be in part responsible by TLR7/8 activation through recognition of activating DAMPs.

Taken together, we suggest that activating TLR7 and TLR8 represents a potential therapeutic treatment that could enhance viral immunity and clearance. Imiquimod is a dual TLR7/8 agonist, which has been suggested as a potential pharmaceutical treatment for COVID-19 patients [182]. This hypothesis is further supported by results demonstrating suppression of inflammation and viral replication in murine models treated with imiquimod after influenza A infection [183]. After established influenza A infection, direct delivery of imiquimod into the lungs (through intranasal administration, but not topical application) was able to reduce viral replication, prevent pulmonary inflammation and leukocyte infiltration, protect against worsening of pulmonary dysfunction, and increase concentration of pulmonary immunoglobulins, resulting in an accelerated recovery [183]. In this study, 8–12-week-old male C57BL6/J mice were infected with the influenza A virus 24 h prior to a two-day treatment of imiquimod (50 μg; administered once daily) [183]. Treatment with imiquimod prevented significant weight loss associated with influenza A infection (reaching a maximum of 6% at day 4, when compared to virally infected mice at 15% at day 5), with mice fully recovering by day 10, three days earlier than non-treated mice [183]. Imiquimod was also able to reduce viral titers and suppress pulmonary inflammation (~40%), demonstrated by significantly reduced neutrophil (~50%) and eosinophil (~70%) counts, and neutrophil chemotactic cytokines (i.e., IL-1β and -6, CCL3, and CXCL2) [183]. Suppression of peri-bronchiolar inflammation and immune cell infiltration protected pulmonary tissue from increased dysfunction, as shown by no significant changes in respiratory resistance, and tissue hysteresivity and damping, when compared to non-treated mice [183]. Finally, imiquimod treatment yielded a significant increase in bronchiole fluid antibodies (i.e., IgG1, IgG2a, IgE, and IgM), indicating a more potent local antibody response critical for aiding in the clearance of viral infection [183]. Additionally, imiquimod may be a potential adjuvant to be incorporated into a SARS-CoV-2 vaccine, due to its ability to augment production of the antigen specific antibody response [184,185]. BALB/c peritoneal B cells incubated with imiquimod (50 μg) and inactivated H1N1/415742Md virus particle (10 μg) resulted in increased B cell proliferation and differentiation, and augmented production of viral neutralising antibodies (i.e., v-IgM and v-IgG) [184,185]. Additionally, when administered as a intraperitoneal injection to 6–8-week-old famle BALB/c mice, increased spleen and mesenteric lymph node B cell numbers and activation were reported within 18 h post injection [184]. Furthermore, mice vaccinated and then challenged with active influenza virus had significantly higher B cells counts in the spleen and mediastinal lymph nodes and higher levels of viral specific IgA in bronchiolar fluid by day 3 post infection [184]. Thus, imiquimod has the potential to be used as both a treatment for COVID-19 patients with established infection, and as an adjuvant in a SARS-CoV-2 vaccine to prime and strengthen the immune response for accelerated viral clearance.

## 4. C-Lectin Type Receptors Involved in COVID-19

### 4.1. Introduction to the C-Lectin Type Receptors

CLRs are a family of transmembrane and soluble PRRs that contain one or more homologous carbohydrate-recognition domains [186,187], allowing for calcium-dependent recognition of glycosylation signatures present on pathogens or host-derived proteins [35]. CLRs can be subdivided into two groups: (i) Mannose receptor family (MR) and (ii) asialoglycoprotein receptor family (i.e., DC-SIGN; L-SIGN; MGL) [188]. CLRs identify pathogens by interacting with mannose, fucose, and glucan mono- and polysaccharide structures [189], which orchestrates viral [190], fungal [191], and mycobacterial [192] immunity. Recognition of PAMPs by CLRs results in pathogen-uptake, degradation, and subsequent antigen presentation [193]. CLRs are able to function independently to induce NF- κβ-mediated inflammation through immunoreceptor tyrosine-based activation motifs and enzymatic activity of spleen tyrosine kinase [194]. Additionally, CLRs are able to cross-communicate and couple with other PRRs (including TLRs) [194,195], allowing for strengthened or dampened innate immune system inflammatory responses by augmenting and abrogating receptor activation and signal transduction [194]. The unique antigen presenting ability displayed by CLRs enables modulation of differentiating adaptive immune cells, and induces pathogen-tailored adaptive immune responses [187]. In vitro models have emerged demonstrating direct association between selective CLRs and SARS-CoV-2 S protein (MR; DC-SIGN; L-SIGN and MGL) mannosylated and N- and O-glycans [34,35], highlighting them as strategic therapeutic targets for the treatment of COVID-19. Furthermore, additional CLRs that may have other roles in COVID-19 (BDCA-2; CLEC2; Dectin-1 and -2; DCIR; DNGR1; LOX-1; LSECtin) [196,197,198,199,200,201,202,203], such as contributing to immunopathological manifestations and severe symptoms experienced by COVID-19 patients, will also be explored.

### 4.2. Blood Dendritic Cell Antigen-2

BDCA-2 is a type II CLR [204], which is coupled with an immune receptor tyrosine-based activation motif (ITAM) located on the intracellular tail [205]. When the ITAM on BDCA-2 is stimulated it recruits the enzyme spleen tyrosine kinase [205]. Phosphorylation of spleen tyrosine kinase results in its activation, which leads to the inhibition of the NF-KB signaling pathway, and TLR9-activated type I IFNs and cytokines on tumor cells [204,206]. In addition, it was previously demonstrated that the glycoprotein E2, on hepatitis C virus binds to BDCA-2 on plasmacytoid DC cells purified from peripheral blood mononuclear cells, resulting in the inhibition of IFN-alpha production [207]. However, whether the S glycoprotein of SARS-CoV-2 can directly bind to BDCA-2 is yet to be determined. A study demonstrated that patients presenting with chilblain-like lesions on their toes, caused by SARS-CoV-2, expressed plasmacytoid DC with activated BDCA-2 [196]. As such, BDCA-2 might be another potential receptor utilized by SARS-CoV-2 mediating the inflammatory response in COVID-19 patients.

### 4.3. C-Type Lectin-Like Receptor 2

CLEC2 is a CLR II transmembrane receptor [208], expressed on natural cells, DCs, monocytes, granulocytes, platelets, megakaryocytes, and liver sinusoidal endothelial cells [209]. CLEC2 has been identified as an essential platelet-activating receptor, required for homeostasis and thrombosis [208]. CLEC2 has shown to interact with HIV-1, acting as a viral attachment factor [210]. It has been shown that CLEC2 mediated capture and transfer of HIV-1 by palettes, resulting in possible systemic infection [210]. Furthermore, CLEC2 has been shown to play an essential role in various acute viral infections [197]. Interestingly, it was shown that CLEC2 binds sialylated O-glycans [211]. O-glycans are located on the SARS-CoV-2 spike protein (Figure 1) [22]. Therefore, it is possible that CLEC2 could bind O-glycans, mediating SARS-CoV-2 viral entry or spread of infection, by facilitating capture and transfer between platelets.

### 4.4. Dendritic Cell-Associated C-Type Lectin-1

Dectin-1 is a type II transmembrane protein [212] expressed on various innate immune cells (DCs, macrophages, eosinophils, monocytes, neutrophils [209]), which uses a single CRD for the identification of foreign glycosylated moieties present on fungi [213] and mycobacteria [192]. Dectin-1 specifically recognizes beta 1,3- [214] and 1,6-linked [215] glycans present on pathogens, and its activation results in innate inflammatory immune responses; of which, it can associate with TLR2 [216] and TLR4 [217], resulting in receptor synergism and amplification of TLR-induced inflammatory response. A recent study demonstrated that Dectin-1 fails to directly bind to SARS-CoV-2 S protein [198]. However, due to increased expression of TLR4 [100] and activating DAMPs of TLR2 [96] and 4 [94,96,100,128,129,130,131,132,133] in patients with COVID-19, Dectin-1 may participate in cross-communication with TLRs during S protein and DAMP identification and stimulation. Furthermore, due to the ability of Dectin-1 to identify glycosylated motifs it may be able to recognize specific glycans located on the S protein of SARS-CoV-2 and further promote exacerbated inflammatory responses. However, further investigation is required to determine the ability of Dectin-1 to identify viral glycans. Inhibition of Dectin-1 may be an appealing approach to dampen inflammation seen in COVID-19 and increase positive patient outcomes. Thus, severe and lethal immunopathological manifestations in reported in COVID-19 may be in part attributed by amplification of TLR2 and 4 inflammatory pathways through synergism with Dectin-1. Therefore, inhibition of Dectin-1 is an appealing approach to dampen inflammation seen in COVID-19 and increase positive patient outcomes. 

### 4.5. Dendritic Cell-Associated C-Type Lectin-2

The carbohydrate-binding region of Dectin-2 recognizes mannose residues present on fungi and lipopolysaccharides on bacterial pathogens, and results in stimulation of the innate immune response [218]. It has been demonstrated that upon Dectin-2 stimulation, reactive oxygen species and the efflux of potassium ions leads to pyrin domain containing 3 (NLRP3) inflammasome-activated production of pro-inflammatory IL-1beta during *Schistosoma mansoni* infection in mice treated with shistosomal egg antigen [219]. As with DCIR, Dectin-2 also contains a carbohydrate binding domain [220]. Dectin-2 recruits the associated Fc-receptor-g motif, situated on the intracellular tail of Dectin-2, to recognize high-mannose glycans, including Candida albicans (pathogenic yeast) and zymosan (found on the surface of fungi) [220]. The expression of Dectin-2 on the surface of the innate immune cells (neutrophils, macrophages, DC, monocytes [209]) has been noted to be elevated early in inflammatory reactions [221]. Dectin-2 upregulation has been demonstrated in human macrophages infected with MERS-CoV after 24 h, suggesting a potential contribution for Dectin-2 as a receptor in mediating SARS-CoV-2 pro-inflammatory response [199]. Given the ability for Dectin-2 to recognize high-mannose glycans on other cells such as yeast and fungi we believed that Dectin-2 may bind mannose residues located on the SARS-CoV-2 spike protein and therefore mediate viral entry. However contrary to this hypothesis and the binding affinity shown by other CLRs, Dectin-2 does not bind to S or its S1 subunit, indicating that it may not act as a receptor mediating SARS-CoV-2 viral entry. However, Dectin-2 did bind (EBY-100) a yeast extract acting as the positive control [35] This suggests that Dectin-2 may only recognize specifically mannan-expressed pathogens, which has yet to be investigated in SARS-CoV-2 mapping. However, these results do not eliminate Dectin-2 as a possible receptor, as it may recognize and bind to different glycan regions of SARS-CoV-2. However, further investigations are required to determine this hypothesis.

### 4.6. Dendritic Cell Immunoreceptor

DCIRs are predominantly expressed on DCs, macrophages, neutrophils, B cells, and plasmacytoid DC, and contain an immune-tyrosine-based inhibitory motif (ITIM) located on the cytoplasmic tail [222]. Activation of the ITIM, by phosphatases src homology containing protein tyrosine phosphatase 1 and 2 [222], has been linked with inhibition pro-inflammatory cytokines, IL-12, and TNF-alpha, through the TLR8 pathway [223] and IFN-alpha production released from the TLR9 pathway [224] in plasmacytoid DCs. The DCIR is expressed on immature DCs and directly interferes with HIV-1 binding [225]. This interference with HIV-1 promotes T cell infection, resulting in a decrease of cell-mediated immunity and cell death [225]. Previously it was noted that glycans containing fucose and mannose domains specifically bound to DCIR expressed in Chinese hamster ovary cells [226]. The same study demonstrated that the N-glycosylated region within the carbohydrate receptor domain of DCIR has a high affinity for the glycoprotein gp140 on HIV-1 [226]. Since DCIR has distinct affinity for glycosylated (specifically mannose) glycans and binds to viruses through this region, we postulate that SARS-CoV-2 may have affinity for the N-glycosylated center of the DCIRs carbohydrate-binding site. Despite the lack of SARS-CoV2/DCIR interaction in the literature, healthy peripheral blood monocular cells infected with SARS-CoV-2 and SARS-CoV induced plasmacytoid DC activation [200,227]. There was no significant detection of cell surface ACE2 on purified and SARS-CoV-2-activated plasmacytoid DC cells, suggesting that the S protein of SARS-CoV-2 binds to DCIR on plasmacytoid DCs [200]. Interestingly, upon plasmacytoid DC activation by SARS-CoV-2, viral RNA accumulation was negative but IFN-alpha production was elevated [200]. This suggests that plasmacytoid DCs cannot be infected by SARS-CoV-2 and that the DCIR on plasmacytoid DC may help present the virus to T cell for elimination [200]. Therefore, we suggest that DCIR recognizes the S protein of SARS-CoV-2, and may have a role in viral control in the early stages of infection.

### 4.7. Dendritic Cell Natural Killer Lectin Group Receptor-1

DNGR1 is responsible for presenting exogenous antigens, to CD8+ T cells [228]. Patients infected with SARS-CoV-2 have impaired DC and CD8+ T cell responses [201]. Indeed, a clinical retrospective study of patients from Wuhan displayed significantly lower levels of T lymphocytes, correlating with disease severity and mortality [229]. It has been previously noted that DNGR1 may be compromised in HIV and Simian immunodeficiency virus infections, consequently reducing cross-presentation of antigens therefore reducing T cell activity against the virus [230]. This process has been suggested to play a role in the pathogenesis of HIV and acquired immune deficiency syndrome [230]. Considering this, reduced T lymphocytes observed in severe COVID-19 patients may be due to DNGR1. Therefore, DNGR1 may represent a potential target for COVID-19 treatment.

### 4.8. Dendritic Cell-Specific Intracellular Adhesion Molecule-3-Grabbing Non-Integrins and Homologue Dendritic Cell-Specific Intercellular Adhesion Molecule-3-Grabbing Nonintegrin Related

DC-SIGN and L-SIGN are type II C-type lectin receptors, expressed on DC and macrophages [209]. DC-SIGN/L-SIGN ligands are carbohydrate dependent, and recognize cell surface ligands, highly saturated with N-linked- high-mannose oligosaccharides and branched fucosylated carbohydrates [231,232]. L-SIGN and DC-SIGN contain tandem repeats of a 23-amino acid sequence, tailed with a C-terminal C-type CRD, which they use to bind high-mannose oligosaccharides [233]. DC-SIGN and L-SIGN bind recombinant S1, which is involved in ACE2 recognition. It has therefore been suggested that both DC-SIGN and L-SIGN may bind the recombinant S protein via the high-mannose and complex N-glycans (Figure 5).

DC-SIGN was originally identified as an attachment factor for human immunodeficiency virus (HIV), increasing infection through binding of the viral envelope protein [231]. DC-SIGN expression is linked to heightened infection efficiency in a variety of viruses, including HIV [231], Simian Immunodeficiency Virus [234] and Influenza A [235]. DC-SIGN modulates the activation of TLR-induced immune response, which is glycan specific, strengthening the inflammatory response [194,195]. DC-SIGN and L-SIGN have been shown to facilitate transmission of SARS-CoV virus to susceptible cells via ACE2 [231,236]. In the quail-derived cell line QT6, (a non-susceptible cell line to SARS-CoV S-mediated infection) transfected with SARS-CoV S protein, Vesicular stomatitis virus G protein and Ebola virus G protein (EBOV GP), DC-SIGN and L-SIGN expression correlated with enhanced EBOV GP mediated infection. However, it did not influence SARS-CoV facilitated infection. Interestingly, the presence of these lectins in ACE2 transfected cells enhanced infectivity, suggesting that alone DC-SIGN and L-SIGN do not mediate cellular entry. However, they do enhance infection of susceptible cells to SARS-CoV [231]. It has been elucidated that the SARS-CoV-2 S protein used to enter cells is highly glycosylated. It has previously been demonstrated in models of influenza, the efficiency of DC-SIGN mediated infection is glycosylation dependent of the viral hemagglutinin [235]. DC-SIGN has shown to mediate binding of SARS-CoV pseudo-typed vectors to human DC with uptake into endosomes. The virus is then delivered via an infection synapse, which is also observed between DC and T cells in studies investigating DC-SIGN involvement in the pathogenesis of HIV [237]. It has also been observed that infectivity of pseudocytes containing SARS-CoV is affected when glycosylation is reduced, highlighting the role of S protein glycosylation in the pathogenesis of SARS-CoV. This may also be occurring in SARS-CoV-2. There is a strong link between DC-SIGN nucleotide polymorphism and SARS severity [238]. It has been suggested that the SARS-CoV S proteins share a resemblance to the structure of HIV envelope proteins. Given the involvement of DC-SIGN in HIV, it was hypothesized that single nucleotide polymorphisms may be involved in the pathogenesis of SARS-CoV [238]. In the early phases of the pandemic, a SARS-CoV-2 variant in the S protein D614G became dominant throughout the world [239]. It has been suggested that not only is G614 the more dominant variant it is also more infectious than D614. It is suggested that the mutation is present in highly glycosylated S proteins. Replacement of an aliphatic G instead of a polar D by at residue 614 has been suggested to increase glycosylation and, consequently, the severity of SARS-CoV-2. Binding of this G variant to DC- or L-SIGN (Figure 5) in both type II alveolar cells, as well as DCs, could contribute to the increased severity of SARS-CoV-2 [240].

L-SIGN is expressed in type II human alveolar cells and lung endothelial cells, and was previously identified as a receptor for SARS-CoV [241], suggesting that in addition to ACE2 and DC-SIGN, L-SIGN may be a receptor facilitating cellular entry of SARS-CoV2 into cells. The role of L-SIGN has been explored in the SARS-CoV. The results discovered that although L-SIGN is a contributing receptor for viral entry, it does so relatively inadequately. It was suggested that the glycans located on the spike protein of the coronavirus are saturated with high mannose type glycans rather than the other types [242]. Interestingly, L-SIGN is expressed in both SARS-CoV and non-SARS infected lung cells. In comparison to heterozygous L-SIGN cells, cells that express homozygous L-SIGN display increased binding of SARS-CoV, higher proteasome-dependent viral degradation, and a lower capacity for *trans* infection [233]. It has been suggested that L-SIGN mediates trans but not cis infection of SARS-CoV [233]. This was confirmed by ACE2 expressing fresh Vero E6 cells. Cells were incubated for 24-h, with results showing, in cultures of fresh Vero E6 cells with virus-pulsed L-SIGN/CHO, cells had increased total viral copy numbers. This specifies that L-SIGN can transmit SARS-CoV to highly permissible cells in a trans manner. To investigate if L-SIGN can mediate cis- infection, ACE2 and L-SIGN were investigated alone and in combination. Results showed no increase in virus replication when both receptors were present indicating no facilitation of cis infection by L-SIGN in SARS-CoV. Cells transfected into Chinese hamster ovary cells expressed human L-SIGN glycoprotein, and susceptibility to SARS-CoV was increased [241]. Given this information, it is likely that L-SIGN is involved in SARS-CoV2 (Figure 5).

There are currently two pre-print articles yet to be peer reviewed that elucidate the role of DC/L-SIGN in SARS-CoV-2. The affinity of the currently known receptors shown to be involved in SARS-CoV-2 were investigated. Results showed that ACE2 had the highest binding affinity, followed by DC-SIGN. However, L-SIGN had a much lower affinity for SARS-CoV-2 [35]. SARS-CoV-2 S protein interacts with multiple innate immune receptors, as a study showed that DC/L-SIGN both bind to the receptor binding domain of SARS-CoV-2 to mediate entry [34]. The binding of DC-SIGN can initiate internalization of the S protein in 3T3-DC-SIGN+ cells, highlighting its role in cellular entry of viruses. Flow cytometry was used to investigate the internalization of the S protein by DC-SIGN. This showed that after 30 min, DC-SIGN had indicators suggesting internalization of the spike protein had occurred. They concluded that DC-SIGN and L-SIGN can recognize SARS-CoV-2 leading to its internalization stimulating infectivity [35].

Expression of macrophages, monocytes, and DC has been associated with the pathogenic inflammatory response observed in COVID-19 patients. Expression of CLR receptors in healthy people and patients experience moderate and severe COVID-19 symptoms showed that patients with severe COVID-19 had elevated expression of DC-SIGN. Additionally, increased expression of key pro-inflammatory markers including, IL-1beta, IL-6, TNF-alpha, CCL2,CCL3, CXCL8, and CXCL10, was noted [35]. Neutralization assays were performed to investigate L-SIGN mediated SARS-CoV-2 entry. It was noted that soluble L-SIGN-fc neutralized viral entry by 48%. Furthermore, DC/L-SIGN overexpressed in HK-293 cells resulted in infectivity by SARS-CoV-2 promiting viral replication [34]. This potentiates a role for CLRs in the pathogenesis and cellular entry of SARS-CoV-2, as well as a role in inducing immunopatholoical side effects and aggumenting the inflammatory repsonse by cooperating with TLRs.

### 4.9. Lectin-Like Oxidized Low-Density Lipoprotein Receptor-1

LOX-1 is a CLR PRR, expressed on DC [243]. Although, to date, there are no studies implicating LOX-1 as a direct receptor for the SARS-CoV-2 S protein, it has shown to enhance expression of DC, as has been shown with other CLRs (DC-SIGN and L-SIGN) to aid in SARS-CoV-2 cellular entry. Ligand binding to LOX-1 has been shown to trigger intracellular signaling, stimulating cellular processes linked to increased risk of CVD [244], and LOX-1 is involved in internalization of oxidized low-density lipoprotein promoting pro-atherogenesis [245]. In addition, COVID-19 patients in intensive care have been shown to have increased LOX-1 expression compared to those not in intensive care and has been suggested as a plausible characteristic to identify patients at greater risk of thrombosis, a severe and often fatal complication associated with COVID-19 [202]. Despite a lack of research into the role of LOX-1 as a receptor mediating SARS-CoV-2 entry, it is possible that LOX-1 could aid other receptors involved in SARS-CoV-2 viral entry or mediate it itself.

### 4.10. Liver and Lymph Node Sinusoidal Endothelial Cell C-Type Lectin

LSECtin is a glycan-binding receptor [209], which specifically identifies and engages carbohydrate motifs and glycoprotein structures that contain (N-acetyl-) glucosamine, fucose, and mannose, and not those composed of galactose [209,246]. Cellular expression of LSECtin is restricted to hepatic and lymph node sinusoidal endothelial cells, Kupffer cells, DCs, and macrophages [209]. LSECtin is involved in recognition of self- and pathogenic-glycoprotein and carbohydrate signatures present on pathogens and subsequent pro-inflammatory mechanisms [247], antigen capture [248], clearance of apoptotic cells through macrophage engulfment [249], and inhibition of T cell activation [250]. Additionally, LSECtin has been shown to play a role during viral invasion, as it has been demonstrated to directly associate with glycoproteins present on the surface of the S-protein of SARS-CoV [246]. Furthermore, during SARS-CoV infection, LSECtin may act as an attachment factor by binding to N-linked glycosylation sites present on the S-protein, and allow for enhanced viral infection by acting as a co-receptor in conjunction with ACE2 to promote efficient viral entry into host cells [246]. Currently, the role that LSECtin has in human SARS-CoV-2 infection remains elusive. However, studies involving ferrets, intranasally inoculated with SARS-CoV-2, suggest that LSECtin may represent a potential receptor involved in viral infection and transmission [203]. Ferrets are used in models of upper respiratory tract infections and respiratory diseases due to their high vulnerability to a diverse number of viral pathogens [251]. Studies have shown a discrepancy of ACE2 distribution, and a low binding score with SARS-CoV-2 [203]. However, they remain susceptible to infection and can transmit the virus to naïve ferrets [252], suggesting that SARS-CoV-2 may utilize additional receptors for infectivity [203]; of which LSECtin has been proposed as a possible viral entry point, as it has previously been shown to facilitate and augment viral infection [203]. Therefore, LSECtin may represent an alternative receptor employed by SARS-CoV-2 to infect cells through recognition of mannosylated N-glycan and O-glycan moieties present on SARS-CoV-2 S protein [1,22]. However, in vitro and in vivo studies are needed to determine the possibility of this theory.

### 4.11. Macrophage Galactose Type C-Type Lectin

MGL has been suggested as a potential receptor for SARS-CoV-2 cellular entry, and has previously been shown to play a role in viral pathology [253]. MGL expression is high in human lung and upper airway tissue, and localization of MGL expression within these tissues was primarily isolated to DCs and macrophages [253]. It has been shown that MGL expression in patients experiencing severe COVID-19 symptoms is extensive [35]. MGL binding is suggested to be glycan-dependent [35]. MGL identifies glycans bearing terminal Gal or GalNAc residues was shown to interact with the S protein. This illustrates that complex N-glycans could potentially anchor SARS-CoV-2 S protein to cell surface CLRs including MGL [254]. O-glycans located at Thr323 or Ser325 on the spike protein have been suggested to be involved in MGL binding to SARS-CoV-2 [198]. This suggests that while DC-/L-SIGN binds to the S1 through either high-mannose or N-glycans, MGL binding is attributed to both N-glycans and O-glycans [35]. This information and the fact that the S protein of SARS-CoV-2 is highly glycosylated with N-glycans and also with O-glycans [21] could explain the role of MGL in SARS-CoV-2 viral entry and could be a beneficial therapeutic target.

### 4.12. Mannose Receptor

The MR or CD206 is a glycoprotein CLR and is predominantly expressed on DCs and macrophages [209]. The extracellular domains of the MR contain eight CRD, which specifically binds to mannosylated residues [255,256]. The homeostatic functions of MR include clearance of unwanted mannosylated proteins from the circulation and recognition of various microorganisms including viruses and bacteria [40,257]. Although the role of the MR in viral infection is not clear, it has been shown that MR directly binds to SARS-CoV and SARS-CoV-2 S protein, which may be due to the abundance of mannose residues on the S protein (Figure 1) [35]. Using computational chemistry models of the glycoprotein structure of the SARS-CoV-2 S protein, it was noted that the S1 protein consisted of N-glycosylation domains highly mannosylated compared to the receptor binding domain, which is the main site utilized for ACE2 interaction [258]. Further ZDOCK and PDBePISA (Proteins, Interfaces, Structures, and Assemblies) interface interaction analysis showed that the mannose presented at the N-glycosylation positions interacts with MR at the N-terminal domain of the S1 protein [258]. It was further demonstrated that the highly mannosylated regions of the S protein were recognized with a strong affinity by the CRD of the MR in HEK292 cells [35]. Moreover, in vivo studies involving human monocytes have shown a cooperation between TLR2, TLR4, and MR during fungal infection [259]. Receptor synergism between MR and TLR2 and 4 may account for the severe inflammation and resulting immunopathological consequences that are reported in patients infected with SARS-CoV-2 (Figure 5). Taken together, the data reveal that the level of COVID-19 severity by SARS-CoV-2 may be in part due to the heavily glycated mannose regions of the S protein and its binding to MR and synergism with TLRs. As such, targeting the mannosylated regions of the S protein could be a potential therapeutic target for patients with severe COVID-19.

## 5. Other Immune Receptors That May Participate in SARS-CoV-2 Infection

### 5.1. Dipeptidyl Peptidase-4

DPP4 or CD26 is an ectopeptidase (transmembrane, plasma membrane protein) with extensive research linking its involvement to physiological processes of the immune system [261]. Currently, inhibiting DPP4 is a treatment for type-2 diabetes [261]. DPP4 is a receptor that is suggested to act as a co-receptor for ACE2 to mediate SARS-CoV-2 cellular entry [38]. Therefore, its inhibition may be a beneficial therapeutic target against COVID-19 by reducing entry into host cells [38]. DPP4 may have potential involvement in N-glycan binding in both MERS-CoV and porcine respiratory coronavirus [38,262]. Furthermore, it is a known receptor for HIV infection, implicating it as having a role during viral infection [261]. DPP4 has also been predicted to interact with the S1 of the SARS-CoV-2 S protein [263]. However, this hypothesis has not been confirmed, as in vitro and in vivo studies are required. DPP4 is expressed in a variety of cells and organs, including epithelial and endothelial cells within the vasculature, lungs, kidney, small intestine, and heart [264]. It has been postulated that DPP4 mediates viral entry in the lungs, not relying on additional co-receptors [38]. Thus, it has been suggested as a key contributor to the extreme inflammatory response observed in fatal COVID-19 pneumonia [38]. The use of DPP4 inhibitors is an attractive approach to be investigated in the treatment of COVID-19 patients who have pre-existing comorbidities, as its inhibition may prevent viral entry and subsequent replication in airways, and prevent the cytokine storm associated with severe COVID-19 complications [38,264]. However, as there is lack of data supporting this hypothesis the involvement of DPP4 as a receptor involved in SARS-CoV-2 entry it is impossible to delineate its role [265]. Two reviews recently published have outlined the potential of DPP4 inhibition to be involved in the pathogenesis of COVID-19 rather than a therapeutic option [261,265]. Given this information, it is clear that two opposing theories exist regarding DPP4 in relation to SARS-CoV-2. Thus, further experimental evidence is needed to identify the physiological involvement of the DPP4 receptor in mediating viral entry in order to validate the potential use of DPP4 inhibition as a COVID-19 treatment.

### 5.2. Neuropilin-1

NRP1 is a pleiotropic transmembrane polypeptide [266], which acts as a co-receptor for a plethora of growth factors (i.e., fibroblast growth factor; hepatocyte growth factor; platelet-derived growth factor; transforming growth factor beta; and vascular endothelial growth factor) [267] to facilitate the regulation of biological processes, including angiogenesis, guidance of axons [268,269], ganglion genesis [269], and vascular permeability [270]. Recent literature has established NRP1 as a host receptor that mediates cellular entry and infectivity of SARS-CoV-2 [266,271]. SARS-CoV-2 contains a polybasic cleavage site (RRAR) between the S1 and S2 spike protein subunits [271]. This site enables cleavage by furin in addition to other proteases, and has been suggested to play an influential role for viral infectivity of cells and increase SARS-CoV-2 pathogenicity [271]. NRP1 is located on the cell surface is subsequently activated by RRAR cleavage by furin [271]. The SARS-CoV-2 spike protein contains (RXXROH) a furin cleavage motif within the S1/S2 junction cleavage site [272]. Studies in HEK-293T cells that express non-detectable ACE2 or NRP1 transcripts were transfected with NRP1, ACE2, and TMPRSS2 (the two key host factors required for viral entry) [272]. The results of this study showed that ACE2 alone increased cell susceptibility to viral infectivity, while NRP1 did not [272]. However, co-expression of cells with ACE2, TMPRSS2, and NRP1 significantly increased infectivity [271]. This suggests that NRP1 may not mediate viral entry, but plays a role in enhancing infectivity in the presence of other host factors, such as ACE2 [271]. However, further studies are needed to determine if the cleavage occurring at the S1-S2 junction results in the formation of a C-terminal end sequence containing substrate for NRP1 to facilitate viral entry [271]. Silver particles coated with the TQTNSPRRAR_OH_ sequence peptide determined that NRP1 expressed on HEK-293 cells promoted uptake of the substrate. Similar results were also observed using olfactory neuronal cells, which reported internalization of TQTNSPRRAR_OH_ sequence peptide by NRP1. Thus, NRP1 is able to associate with and internalize SARS-CoV-2 for cellular entry. {Casalino, 2020 #144}.

## 6. Conclusions

ACE2 has been implemented as a fundamental receptor in SARS-CoV-2 entry into host cells leading to COVID-19 in patients [25]. However, evidence of infected cells lacking or expressing low levels of ACE2 has been noted [31,32,33]; thus, supporting our hypothesis that SARS-CoV-2 may utilize other receptors to infect cells, independent of ACE2 expression. To the best of our knowledge, this review article represents novel literature, which compiles additional non-immune and immune receptors that may be utilized by SARS-CoV-2 S protein to gain entry into cells, by direct association with moieties located on the S protein or receptors that may worsen or protect against severe COVID-19 patient outcomes (Table 1). Selective immune system receptors (TLRs, CLRs, and NRP1) and translocated GRP78 represent additional receptors used by SARS-CoV2 S protein to infect intra- and extrapulmonary cells by utilizing carbohydrate and glycan moieties present on its outer surface. Furthermore, we discuss the possible involvement of other CLRs in other events during SARS-CoV-2 infection (e.g., pro-inflammatory pathways, amplification of inflammatory response). As there is yet to be a vaccine or drug approved to prevent or treat COVID-19, we are confident that focusing on these receptors may be the key to dampening severe adverse immunological reactions observed in patients and improving patient outcomes.

## Figures and Tables

**Figure 1 ijms-22-00992-f001:**
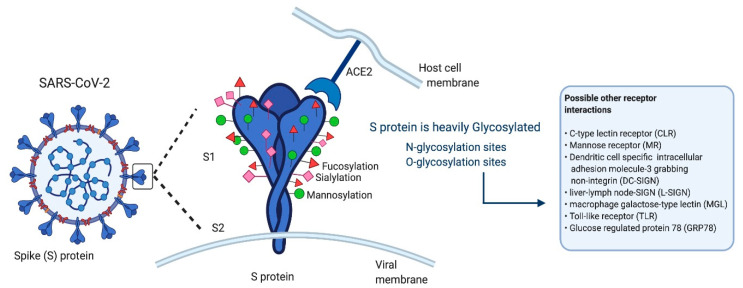
Schematic representation of SARS-CoV-2 virus showing the Spike (S) protein which is heavily glycosylated (O-glycosylation and N-glycosylation) with numerous fucose, mannose and sialyl residues. These sugar moieties have the potential to bind to C-lectin type receptors (CLR), mannose receptor (MR), dendritic cell-specific intracellular adhesion molecule-3-grabbing non-integrin (DC-SIGN), homologue dendritic cell-specific intercellular adhesion molecule-3-grabbing nonintegrin related (L-SIGN), macrophage galactose-type lectin (MGL), toll-like receptors (TLR), and glucose regulated protein 78 (GRP78).

**Figure 2 ijms-22-00992-f002:**
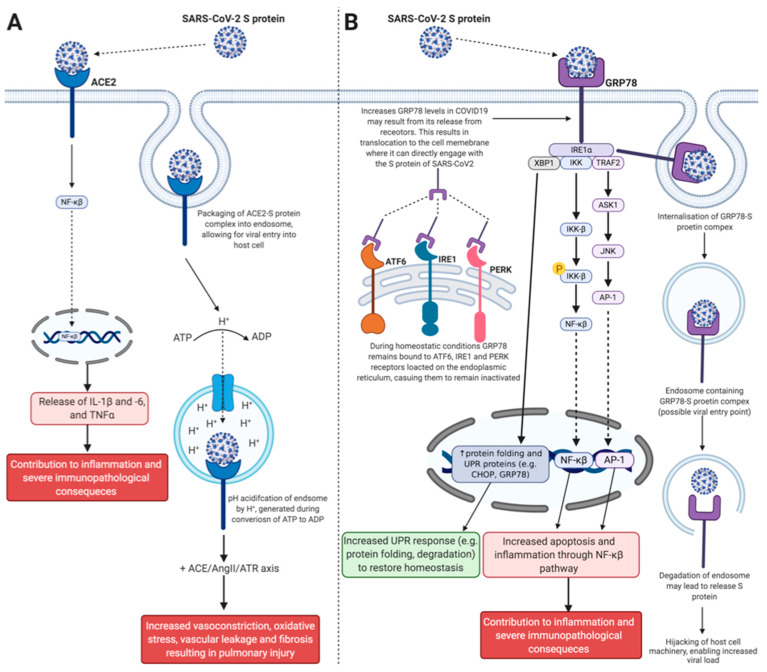
(**A**) Proposed involvement of angiotensin-converting enzyme 2 in SARS-CoV-2 infection. The receptor binding motif of S protein interacts with ACE2 [25], resulting in endocytosis by host cells [26]. Patients with severe COVID-19 have increased levels of IL-1beta, IL-6, and TNF-alpha [45,46], which may be released by NF-kB driven activation, resulting in contribution to immunopathological outcomes [45,46]. Up-regulation of ACE/AngII/AngII type 1 receptor axis may also result in vasoconstriction and pulmonary lung injury [41]. (**B**) Involvement of glucose-regulated protein 78 (GRP78) in SARS-CoV-2 infection. Under homeostatic condition GRP78 remains bound to ATF6, IRE1, and PERK in the endoplasmic reticulum [36]. Increased GRP78 levels have been reported in COVID-19 patients [47,48], suggesting that GRP78 is liberate from its receptors, and translocates to the cell membrane. In fact, GRP78 directly interacts with SARS-CoV-2 S protein [36]. Coupled with the ability of GRP78 to bind to and internalize viral fragments [49], we propose that GRP78 may present a viral entry point exploited by SARS-CoV-2 virus. Abbreviations: activator protein-1, AP-1; adenosine diphosphate, ADP; adenosine triphosphate, ATP; apoptosis signal-regulating kinase 1, ASK1; activating transcription factor 6, ATF6; C/EBP homologous protein, CHOP; hydrogen ion, H+; inhibitor of nuclear factor kappa-beta kinase subunit, IKK; inositol-requiring enzyme-1, IRE1; Jun N-terminal kinase, JNK; X-box-binding protein 1, XBP1.

**Figure 3 ijms-22-00992-f003:**
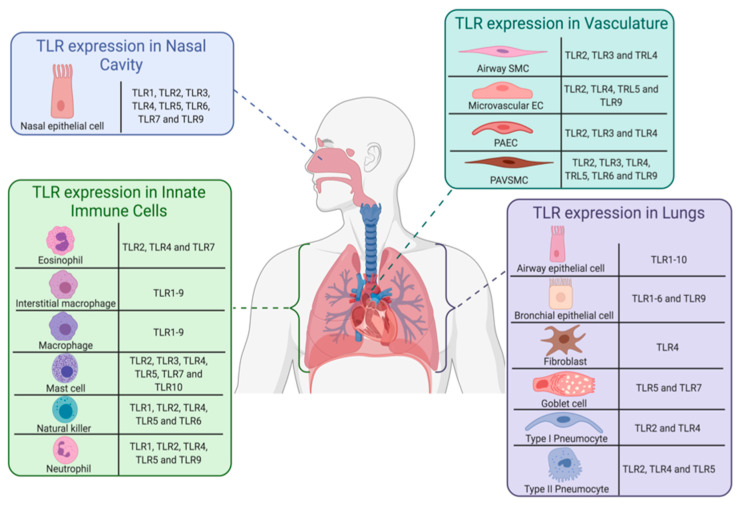
Expression of functional toll-like receptors in specific cell populations of the human respiratory system. Functional expression of TLR1-10 has been reported in human pulmonary tissue [57]. However, there are limited studies available investigating TLR expression in specific cell populations within human respiratory tissue. This illustration depicts expression of TLRs in the nasal cavity (TLR1-7 and TLR9 [67,68]) and specific cell populations in located pulmonary tissue, including innate immune cells (eosinophils: TLR2, TLR4, and TLR7 [69,70]; interstitial macrophages: TLR1-9 [71]; macrophages: TLR1-9 [71,72]; mast cells: TLR2-5, TLR7, and TLR10 [73]; natural killer cells: TLR1, TLR2, TLR4, TLR5, and TLR6 [74]; and neutrophils TLR1, TLR2, TLR4, TLR5, and TLR9 [75]), vasculature (airway SMCs: TLR2-4 [76]; microvascular ECs: TLR2, TLR4, TLR5 and TLR9 [77]; PAECs: TLR2-4 [78,79,80]; and PAVSMCs: TLR2-6 and TLR9 [77,81]) and lung cells (airway epithelial cells: TLR1-10 [82,83]; bronchial epithelial cells: TLR1-6 and 9 [83]; fibroblasts: TLR4 [84]; goblet cells: TLR5 and TLR7 [70]; type I pneumocytes: TLR2 and TLR 4 [85]; and type II pneumocytes: TLR2, TLR4, and TLR5 [70,85]). These cells may contribute to severe immunopathological manifestations experienced by COVID-19 patients, and may be novel entry points used by SARS-CoV-2 for host cell infection. Abbreviations: Endothelial cell, EC; pulmonary artery endothelial cell, PAEC; pulmonary artery vascular smooth muscle cell, PAVSMC.

**Figure 4 ijms-22-00992-f004:**
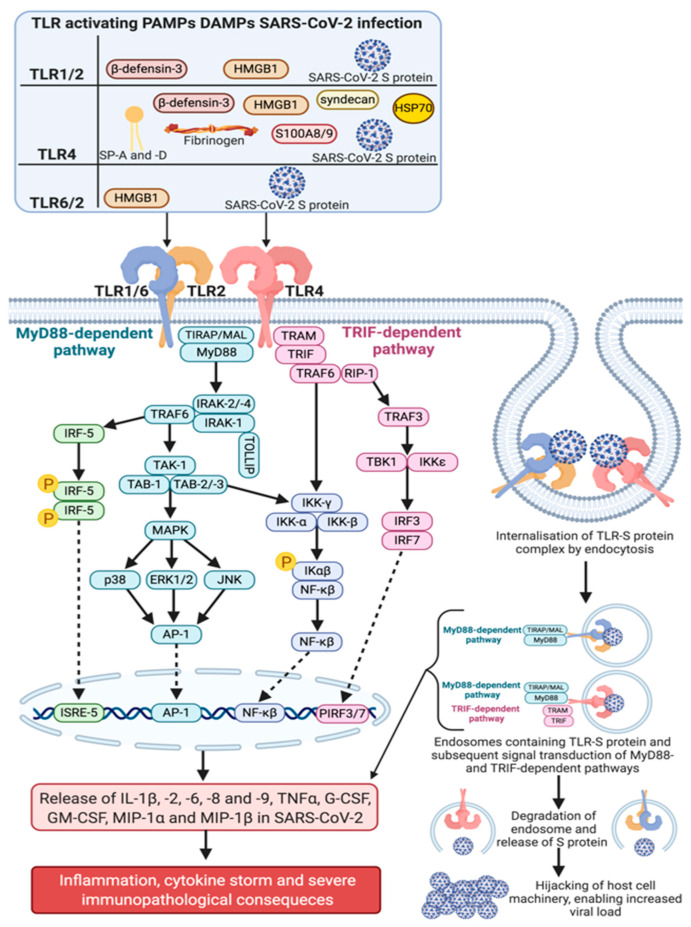
Proposed toll-like receptor 1, 4, and 6 involvement in SARS-CoV-2 infection into host cells. In silico studies have demonstrated direct binding between TLR1, 4, and 6 and the S1 interface of SARS-CoV-2 S protein [2], indicating that it may be a novel TLR pattern recognition receptor. Additionally, circulating TLR activating danger-associated molecular patterns (DAMP) have been reported in COVID-19 patients [94,96,100,128,129,130,131,132,133]. Activation by SARS-CoV-2 S protein and DAMPs may drive TLR mediated inflammation (through myeloid differentiation factor-88- and toll/IL-1-domain-containing adapter-inducing interferon-beta (TRIF)-dependent pathways) in patients with COVID-19 [144], manifesting as chronic inflammation, cytokine storm, and severe outcomes [100]. As such, we postulate that receptor-dependent internalization of the S protein may provide novel viral entry points, causing systemic spread independent of ACE2. *Abbreviations:* activator protein-1, AP-1; extracellular signal-regulated kinase, ERK; granulocyte colony-stimulating factor, G-CSF; granulocyte-macrophage colony-stimulating factor, GM-CSF; k protein 70, HSP70; inhibitor of nuclear factor kappa-beta kinase subunit, IKK; inositol-requiring enzyme-1 alpha, IRE1α; interferon regulatory factor, IRF; interferon stimulated response element, ISRE; interluikin-1 associated receptor kinase, IRAK; c-Jun N-terminal kinase, JNK; macrophage inflammatory protein-1 MIP-1; mitogen-activated protein kinase, MAPK; myeloid differentiation factor-88 adaptor-like, MAL; receptor interacting protein-1, RIP-1; surfactant protein, SP; TANK-binding kinase 1, TBK1; toll/interluikin-1 receptor domain-containing adapter protein, TIRAP; toll/interluikin-1 receptor domain-containing adaptor-inducing interferon-beta, TRIF; toll-like receptor, TLR; toll interacting protein, TOLLIP; transforming growth factor beta activated kinase, TAB; translocating chain-associated membrane protein, TRAM; tumor necrosis factor receptor-associated factor, TRAF.

**Figure 5 ijms-22-00992-f005:**
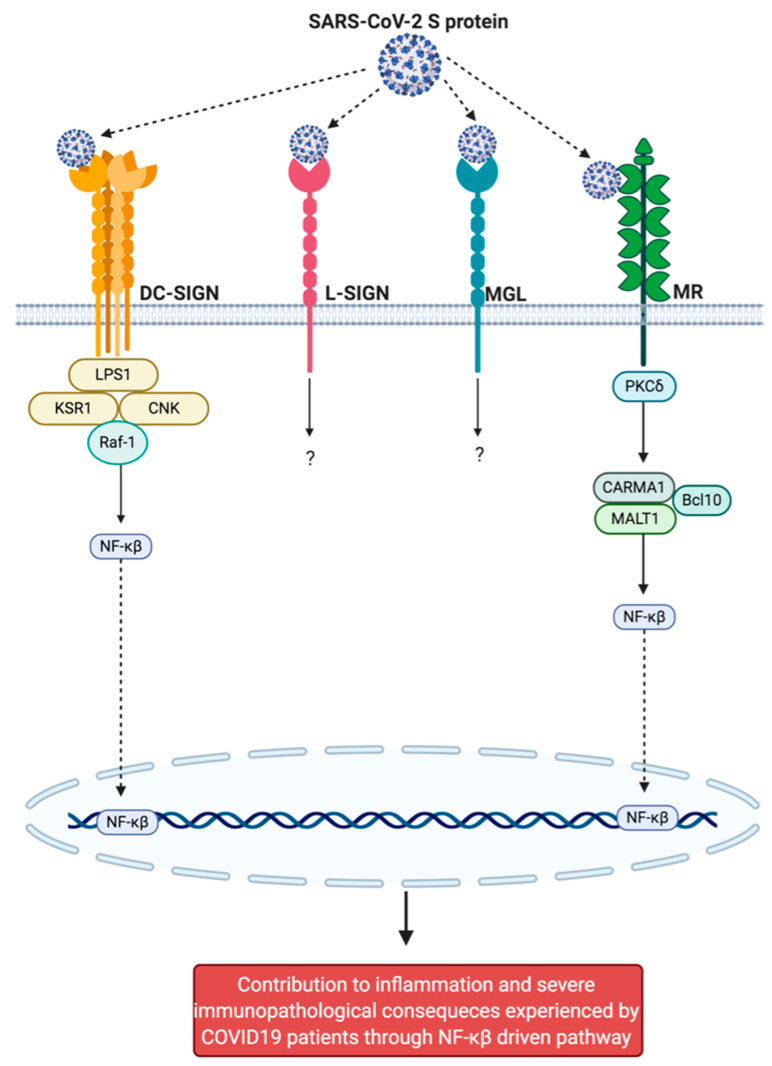
Postulated pathways involving specific C-lectin type receptors (CLR) that have been shown to bind to SARS-CoV-2 spike protein. DC-SIGN, L-SIGN, MGL, and MR have been shown to directly associate with carbohydrate motifs present on the outer surface of SARS-CoV-2 S protein [35]. The signal transduction of L-SIGN and MGL remains undetermined. However, we postulate that DC-SIGN and MR may contribute to inflammation and severe immunopathological manifestations experienced in COVID-19 patients through NF-kB pathway, which has previously been reported in fungal infections [260]. *Abbreviations*: B-cell lymphoma/leukemia 10, Bcl10; caspase recruitment domain-containing membrane- associated guanylate kinase protein-1, CARMA1; connector enhancer of kinase suppressor of Ras1, CNK; kinase suppressor of Ras1, KSR1; lymphocyte-specific protein kinase-1, LSP1; mucosa-associated lymphoid tissue lymphoma translocation protein 1, MALT1; protein kinase C delta, PCKδ; proto-oncogene, serine/threonine kinase, Raf-1.

**Table 1 ijms-22-00992-t001:** Potential novel viral entry points through direct binding of SARS-CoV-2 spike protein and their corresponding mechanism of action during SARS-CoV-2 recognition.

Receptor	Mechanism of Action	Reference
ACE2	S protein receptor binding motif binds to the N-terminal extracellular catalytic ectodomain of ACE2	[25]
DC-SIGN	Receptor binding domain of SARS-CoV-2 S protein	[35]
GRP78	III and IV cyclic regions of S protein	[36]
L-SIGN	Receptor binding domain of SARS-CoV-2 S protein	[35]
MGL	N- and O-glycans present on the S1 of the S protein	[35]
MR	Mannose at N-glycosylation positions at N-terminal domain present on the S1 of S protein	[258]
	Internalization of SARS-CoV-2 substrate	
NRP1		[266,271]
TLR1/4/6	Hydrogen bonding and hydrophobic interactions with S1 of S protein	[2]

## Data Availability

Not applicable.

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
