# Peer review of "Can SARS-CoV-2 Virus Use Multiple Receptors to Enter Host Cells?"

_ijms, 2021, doi:10.3390/ijms22030992_

Round 1

Reviewer 1 Report

Can SARS-CoV-2 virus use multiple receptors to enter host cells?

Manuscript ID: ijms-995924

The manuscript submitted by Gadanec et al. deals with the impact of various immune and non-immune receptors to function as potent entry receptor or at least as a receptor that facilitates or support viral entry. Dividing the review into non-immune (e.g. ACE2) and immune receptors (e.g. TLR, CLR) the authors do not only describe fundamental receptors for SARS-CoV-2 entry (e.g. ACE2) but also receptors that may facilitate viral entry (e.g. DC-SIGN) or protect against severe clinical outcomes in COVID-19 patients (e.g. TLRs). There is an extensive general description of each single receptor before the contribution in viral infections and the impact on SARS-CoV-2 infection is discussed.

Major comments:

  1. Page 4, line 135-138: This sentence came out of nowhere and a link between ACE2 and the stimulation of different factors is somehow missing. Re-writing the sentence may help to make the statement clear.

  1. Page 8, line 300: The authors describe TLR1/2 and TLR6/2 as potential therapeutic target to prevent excessive inflammation and viral entry. Nevertheless, giving a TLR1/2 or TLR6/2 inhibitor to block receptor function time point of administration has to be determined precisely to avoid loss-of-immune function and still prevent inflammation and viral entry. May authors could give a brief statement to this fact.

  1. Page 12, chapter 3.6: The authors describe general aspects of the receptors and mentioned than only one study that represents a bioinformatics analysis with SARS-CoV-2 before making a suggestion how TLR7/8 are involved in SARS-CoV-2 infection. The impact of TLR7 and TLR8 is extensively described for other viruses (e.g. HIV, DENV, Influenza A). Maybe, the authors could link the knowledge about TLR7/8 and other viruses to the findings from bioinformatics analysis.

  1. Page 13, line 489: There might be better references to confirm the influence in pathogenicity than Ref. 169.

  1. Page 13, line 503-505: Does TLR7/8 recognize alpha-Phospholipid antibodies? If so, please integrate one sentence with refence (e.g. Döring et al. 2010 or Prinz et al., 2011)

  1. Page 13, lin e 509-512: Since SARS-CoV-2 is known to induce cytokine storm that leads to severe clinical course of infection, it is important to make a statement about the time point for administration of imiquimod to avoid further induction /secretion of cytokines and chemokines and cellular damage.

  1. Page 14, line 578-584: Description of the receptor function is from yeast/fungi or parasits. Maybe the authors can sum this up.

  1. Even if published quite recently, it would improve the manuscript/review if the authors can add a chapter about neuropilin-1 as host factor for SARS-CoV-2 (https://science.sciencemag.org/content/early/2020/10/19/science.abd3072.long; https://science.sciencemag.org/content/early/2020/10/19/science.abd2985.long)

Minor comments:

  1. Page 3, Line 132: “…patients infected with COVID-19 have…” – you can’t be infected by a disease but only by a pathogen (e.g. SARS-CoV-2)

  1. Page 6, line 215: not only the innate immune system is a host defense system. Also the adaptive immunity is host defense. Maybe skip the part “also known as the host defense system

  1. Page 6, line 227: toll/IL--1 – toll/IL-1

  1. Page 13, line 512: Punctuation mark is missing.

  1. Page 15, line 611: “There was no significant detection of ACE2 antibody in the purified and SARS-CoV-2-activated plasmacytoid DC cells,…” – there can’t be a detection of ACE2 antibodies in the plasmacytoid DCs. In the original paper (reference 191) the authors used ACE2 specific antibodies to detect ACE2 protein in DCs. The sentence needs correction.

  1. Page 16, line 663: “SARS-CoV” – SARS-CoV-2

  1. Page 17, line 705: “was notrd” – was noted

As SARS-CoV-2 is a “young” virus and knowledge is limited, many publications only available as pre-print versions so far. The authors try to avoid the use of this kind of source (15 pre-prints out of 258 references), nevertheless, the authors should recheck, if some reference are already peer-reviewed and accepted for publication.

Author Response

Major comments:

Page 4, line 135-138: This sentence came out of nowhere and a link between ACE2 and the stimulation of different factors is somehow missing. Re-writing the sentence may help to make the statement clear.

Thank you for bringing this to our attention. We agree that this statement was not necessary and has been removed, the link between angiotensin II and ACE2 is discussed further in the manuscript. ACE2 downregulation which is observed in covid-19 means there is reduction in angiotensin I and II degradation leading to increased Angiotensin concentration.

Page 8, line 300: The authors describe TLR1/2 and TLR6/2 as potential therapeutic target to prevent excessive inflammation and viral entry. Nevertheless, giving a TLR1/2 or TLR6/2 inhibitor to block receptor function time point of administration has to be determined precisely to avoid loss-of-immune function and still prevent inflammation and viral entry. May authors could give a brief statement to this fact.

Thank you for pointing this out. To justify the use of TLR1/2 and TLR6/2 inhibitors we have further discussed their use in human cell lines infected with different bacteria to determine when they were given. As there is a gap in the literature pertaining to their use in in vivo models we have discussed the lack of studies and the need to determine when they should be administered (i.e. before or during infection). The paragraph now reads as:

Line 298-314: “The recent discovery of MMG11 (a TLR2 inhibitor, which shows preference for the TLR1/2 heterodimer [1, 2]) and CuCpt22 (a TLR1/2 heterodimer inhibitor in mice [1] and a TLR1/2/6 inhibitor in humans [1, 3]) may represent potential COVID-19 therapeutics. Pre-treatment of MMG11 (5 g/ml) followed by infection with Mycobacterium avium subspecies paratuberculosis in human macrophages, resulted in significantly reduced concentration of pro-inflammatory cytokines IL-8 and TNF [2]. Similar anti-inflammatory abilities have been observed in human primary bronchial epithelial cells pre-treated with CuCpt22 (50 M), 30 minutes before challenge with Streptococcus pneumonia strain D39 [3]. Pre-treatment with CuCpt22 was able to reduce gene expression of IL-6 and granulocyte-macrophage colony-stimulating factor and lowered expression of nuclear factor kappaB inhibitor- [3] (an essential regulator of the TLR response, with increased expression being associated with pulmonary pathologies caused by exacerbated and unregulated inflammation [4, 5]). However, in vitro and in vivo studies investigating the ability of MMG11 and CuCpt22 to be administered as either a prophylactic or treatment during active SARS-CoV-2 infection are required to determine: (a) the optimal time point for the greatest beneficial effect; (b) the ability to prevent viral entry into cells; and (c) the extent to which they can dampen the inflammatory response”

Page 12, chapter 3.6: The authors describe general aspects of the receptors and mentioned than only one study that represents a bioinformatics analysis with SARS-CoV-2 before making a suggestion how TLR7/8 are involved in SARS-CoV-2 infection. The impact of TLR7 and TLR8 is extensively described for other viruses (e.g. HIV, DENV, Influenza A). Maybe, the authors could link the knowledge about TLR7/8 and other viruses to the findings from bioinformatics analysis.

Thank you for this comment, we have provided a more conclusive link as to why TLR7/8 may be responsible, at least in part, for antiviral immunity in SARS-CoV-2, and the paragraph has been corrected to:

Line 497-515: “The ability of TLR7/8 to reduce replication of viruses has been demonstrated in HIV-1 [1], influenza [2] and MERS-CoV [178], as upon entry into the cell viral ssRNA binds to TLR7/8 promoting activation and antiviral immunity. Activation of TLR7 [166, 171] and TLR8 [172] induces the recruitment of the adaptor molecule MyD88, resulting in the release of pro-inflammatory cytokines and chemokines [73], and type I (IFN-alpha and IFN-beta) and III IFNs (IFN-lambda) [173], which have been shown to aid in viral clearance and reduced replication. It remains unknown if TLR7/8 can directly interact with SARS-CoV-2 S protein, upon entry into host cells. However, they have been suggested as possible SARS-CoV-2 therapeutic targets due their anti-viral immunity and ability to sense ssRNA. Recognition of viral genomic ssRNA from positive-sense RNA viruses has been shown to be recognized by endosomal TLR7/8 [174-176]. A bioinformatic analysis, investigating genomic ssRNA fragments of SARS-CoV-2, reported a larger number of fragments (greater than that shown in SARS-CoV) that could be identified by TLR7/8 [177]. These results suggest that rapid release of type I IFNs by TLR7/8 could influence pathogenicity of SARS-CoV-2 by altering: DC growth, maturation and apoptosis, cytotoxicity of natural killer cells, and virus-specific cytotoxic responses produced by T lymphocytes [177]. Additionally, during adenovirus type 5 infection the TLR7/MyD88 pathway was responsible for subsequent signal transduction by lung epithelial cells, necessary for IFN production [178]. Thus, recognition of SARS-CoV-2 ssRNA by TLR7/8 may result in antiviral immunity through increased production of cytokines and IFNs.”

Page 13, line 489: There might be better references to confirm the influence in pathogenicity than Ref. 169.

Thank you for this suggestion, we agree that other references should be used as evidence that ssRNA are agonistic ligands of TLR7/8. To correct this we have added the following three references:

Lund, J.M., et al., Recognition of single-stranded RNA viruses by Toll-like receptor 7. Proceedings of the National Academy of Sciences, 2004. 101(15): p. 5598-5603.

Zhang, Z., et al., Structural analyses of Toll-like receptor 7 reveal detailed RNA sequence specificity and recognition mechanism of agonistic ligands. Cell Reports, 2018. 25(12): p. 3371-3381. e5.

Tanji, H., et al., Toll-like receptor 8 senses degradation products of single-stranded RNA. Nature structural & molecular biology, 2015. 22(2): p. 109.

Page 13, line 503-505: Does TLR7/8 recognize alpha-Phospholipid antibodies? If so, please integrate one sentence with refence (e.g. Döring et al. 2010 or Prinz et al., 2011)

Thank you for this comment, as phospholipid antibodies could induce unwanted inflammation we have expanded on our previous findings to further discuss the role that phospholipid antibodies have in COVID-19, and have included the mentioned references. However, to the best of our knowledge we cannot find literature pertaining to alpha-phospholipid antibodies. The paragraph now reads as:

Line 520-537: “However, due to the simultaneous release of pro-inflammatory cytokines and chemokines, activation of TLR7/8 during SARS-CoV-2 may provoke an augmented inflammatory response, which could result in severe and potentially lethal immunopathological consequences experienced by COVID-19 patients [169]. Patients with COVID-19 have shown increased circulating levels of pro-inflammatory cytokines and chemokines, which are produced through the TLR7/8 pathways [73]. This may be due to TLR7/8 recognizing antiphospholipid antibodies (aPL) (a TLR7/8 activating DAMP [172-174]), which have been shown to be upregulated in COVID-19 patients [175, 176]. aPLs are a family of autoantibodies that associate with negatively charged phospholipids, resulting in the disruption of self-tolerance and launch of autoimmune responses targeting host phospholipids [172-174]. A study investigating the presence of aPLs in severe and critical COVID-19 patients (admitted to the intensive care unit) determined that patients infected with SARS-CoV-2 had increased concentrations of circulating aPLs, when compared to healthy individuals [175]. Of the 21 COVID-19 patients recruited, at least 12 patients had increased circulating levels of at least one aPL (antiannexin V IgM: 19%; anticardiolipin IgM: 14%; antiphosphatidylserine IgM: 14% anticardiolipin IgG: 10%; and antiphosphatidylserine IgG: 10%) [175]. Thus, the exacerbated immune response resulting in cytokine storm may be in part responsible by TLR7/8 activation through recognition of activating DAMPs.”

Page 13, line 509-512: Since SARS-CoV-2 is known to induce cytokine storm that leads to severe clinical course of infection, it is important to make a statement about the time point for administration of imiquimod to avoid further induction /secretion of cytokines and chemokines and cellular damage.

Upon further reading, we agree with this comment as we have failed to provide evidence as to when imiquimod could be administered. Therefore, we have re-written this paragraph explaining how imiquimod could be used as both a treatment once infected and as a adjuvant in a vaccine. The paragraph now reads as:

Line 538-571: Taken together, we suggest that activating TLR7 and TLR8 represents a potential therapeutic treatment that could enhance viral immunity and clearance. Imiquimod is a dual TLR7/8 agonist, which has been suggested as a potential pharmaceutical treatment for COVID-19 patients [177]. This hypothesis is further supported by results demonstrating suppression of inflammation and viral replication in murine models treated with imiquimod after influenza A infection [178]. After established influenza A infection, direct delivery of imiquimod into the lungs (through intranasal administration, but not topical application) was able to reduce viral replication, prevent pulmonary inflammation and leukocyte infiltration, protected against worsening of pulmonary dysfunction and increased concentration of pulmonary immunoglobulins, resulting in an accelerated recovery [178]. In this study, 8-12-week-old male C57BL6/J mice were infected with the influenza A virus 24 hours prior to a two-day treatment of imiquimod (50 g; administered once daily) [178]. Treatment with imiquimod prevented significant weight loss associated with influenza A infection (reaching a maximum of 6% at day 4, when compared to virally infected mice at 15% at day 5), with mice fully recovering by day 10, three days earlier than non-treated mice [178]. Imiquimod was also able to reduce viral titers and suppress pulmonary inflammation (~40%), demonstrated by significantly reduced neutrophil (~50%) and eosinophil (~70%) counts, and neutrophil chemotactic cytokines (i.e. IL-1 and -6, CCL3, and CXCL2) [178]. Suppression of peri-bronchiolar inflammation and immune cell infiltration protected pulmonary tissue from increased dysfunction, as shown by no significant changes in respiratory resistance, and tissue hysteresivity and damping, when compared to non-treated mice [178]. Finally, imiquimod treatment yielded a significant increase in bronchiole fluid antibodies (i.e. IgG1, IgG2a, IgE and IgM), indicating a more potent local antibody response critical for aiding in the clearance of viral infection [178]. Additionally, imiquimod may be a potential adjuvant to be incorporated into a SARS-CoV-2 vaccine, due to its ability to augment production of the antigen specific antibody response [179, 180]. BALB/c peritoneal B cells incubated with imiquimod (50 g) and inactivated H1N1/415742Md virus particle (10 g) resulted in increased B cell proliferation and differentiation, and augmented production of viral neutralising antibodies (i.e. v-IgM and v-IgG) [179, 180]. Additionally, when administered as a intraperitoneal injection to 6-8-week-old famle BALB/c mice, increased spleen and mesenteric lymph node B cell numbers and activation were rported within 18 hours post injection [179]. Furthermore, mice vaccinated and then challenged with active influenza virus had significantly higher B cells counts in the spleen and mediastinal lymph nodes and higher levels of viral specific IgA in bronchiolar fluid by day 3 post infection [179]. Thus, imiquimod has the potential to be used as both a treatment for COVID-19 patients with established infection, and as an adjuvant in a SARS-CoV-2 vaccine to prime and strengthen the immune response for accelerated viral clearance.”

Page 14, line 578-584: Description of the receptor function is from yeast/fungi or parasits. Maybe the authors can sum this up.

Thank you for bringing this to our attention and has been changed:

Line 631-639: “Furthermore, due to the ability of Dectin-1 to identify glycosylated motifis it may be able to recognize specific glycans located on the S protein of SARS-CoV-2 and further promote exacerbated inflammatory responses. However, further investigation is required to determine the ability of Dectin-1 to identify viral glycans. Inhibition of Dectin-1 may be an appealing approach to dampen inflammation seen in COVID-19 and increase positive patient outcomes. Thus, severe and lethal immunopathological manifestations in reported in COVID-19 may be in part attributed by amplification of TLR2 and 4 inflammatory pathways through synergism with Dectin-1. Therefore, inhibition of Dectin-1 is an appealing approach to dampen inflammation seen in COVID-19 and increase positive patient outcomes.”

Even if published quite recently, it would improve the manuscript/review if the authors can add a chapter about neuropilin-1 as host factor for SARS-CoV-2 (https://science.sciencemag.org/content/early/2020/10/19/science.abd3072.long; https://science.sciencemag.org/content/early/2020/10/19/science.abd2985.long)

We are extremely appreciative for this insight to adding another receptor that will lead to the strengthening of this paper. We have added a paragraph on NRP1 and included the references mentioned. The paragraph reads as:

Line 902-926: “NRP1 is a pleiotropic transmembrane polypeptide [270], which acts as a co-receptor for a plethora of growth factors (i.e. fibroblast growth factor; hepatocyte growth factor; platelet-derived growth factor; transforming growth factor beta; and vascular endothelial growth factor) [271] to facilitate the regulation of biological processes, including angiogenesis, guidance of axons [272, 273], ganglion genesis [273] and vascular permeability [274]. Recent literature has established NRP1 as a host receptor that mediates cellular entry and infectivity of SARS-CoV-2 [270, 275]. SARS-CoV-2 contains a polybasic cleavage site (RRAR) between the S1 and S2 spike protein subunits [275]. This site enables cleavage by furin in addition to other proteases, and has been suggested to play an influential role for viral infectivity of cells and increase SARS-CoV-2 pathogenicity [275]. NRP1 is located on the cell surface is subsequently activated by RRAR cleavage by furin [275]. The SARS-CoV-2 spike protein contains (RXXROH) a furin cleavage motif within the S1/S2 junction cleavage site [276]. Studies in HEK-293T cells that express non-detectable ACE2 or NRP1 transcripts were transfected with NRP1, ACE2 and TMPRSS2 (the two key host factors required for viral entry) [276]. The results of this study showed that ACE2 alone increased cell susceptibility to viral infectivity, while NRP1 did not [276]. However, co-expression of cells with ACE2, TMPRSS2 and NRP1 significantly increased infectivity [275]. This suggests that NRP1 may not mediate viral entry, but plays a role in enhancing infectivity in the presence of other host factors, such as ACE2 [275]. However, further studies are needed to determine if the cleavage occurring at the S1-S2 junction results in the formation of a C-terminal end sequence containing substrate for NRP1 to facilitate viral entry [275]. Silver particles coated with the TQTNSPRRAROH sequence peptide determined that NRP1 expressed on HEK-293 cells promoted uptake of the substrate. Similar results were also observed using olfactory neuronal cells, which reported internalization of TQTNSPRRAROH sequence peptide by NRP1. Thus, NRP1 is able to associate with and internalize SARS-CoV-2 for cellular entry.”

Minor comments:

Page 3, Line 132: “…patients infected with COVID-19 have…” – you can’t be infected by a disease but only by a pathogen (e.g. SARS-CoV-2).

Thank you for bringing this to our attention. We have corrected this and the sentence now reads: At the same time of the abovementioned process, critically ill patients infected with SARS-CoV-2 have shown significantly elevated plasma concentrations of AngII [45].”

Page 6, line 215: not only the innate immune system is a host defense system. Also the adaptive immunity is host defense. Maybe skip the part “also known as the host defense system

This has been omitted and the sentence now reads:

Line 215-216:“The innate immune system facilitates first-line defensive mechanisms against invading pathogens [58,59].”

Page 6, line 227: toll/IL--1 – toll/IL-1

The extra hyphen as been removed, and it now reads toll/IL-1.

Page 13, line 512: Punctuation mark is missing.

The correct punctuation mark has been added.

Page 15, line 611: “There was no significant detection of ACE2 antibody in the purified and SARS-CoV-2-activated plasmacytoid DC cells,…” – there can’t be a detection of ACE2 antibodies in the plasmacytoid DCs. In the original paper (reference 191) the authors used ACE2 specific antibodies to detect ACE2 protein in DCs. The sentence needs correction.

This was altered to suggest that antibody detection of ACE2 was on the cell surface of pDC not within them rather then in the cell. The sentence has been corrected and now reads:

Line 680-682: “There was no significant detection of cell surface ACE2 on purified and SARS-CoV-2-activated plasmacytoid DC cells, suggesting that the S protein of SARS-CoV-2 binds to DCIR on plasmacytoid DCs [202].”

Page 16, line 663: “SARS-CoV” – SARS-CoV-2

This has been corrected.

Page 17, line 705: “was notrd” – was noted

The spelling of noted has been corrected.

As SARS-CoV-2 is a “young” virus and knowledge is limited, many publications only available as pre-print versions so far. The authors try to avoid the use of this kind of source (15 pre-prints out of 258 references), nevertheless, the authors should recheck, if some reference are already peer-reviewed and accepted for publication.

As requested we have rechecked all pre-printed literature and have changed refences of those that have been peer-reviewed and published. These include:

Reference 15 has been updated: Wu, C., et al., Effects of renin-angiotensin inhibition on ACE2 (angiotensin-converting enzyme 2) and TMPRSS2 (transmembrane protease serine 2) expression: insights into COVID-19. Hypertension, 2020. 76(4): p. e29-e30.

Reference 101 has been updated: Sohn, K.M., et al., COVID-19 patients upregulate toll-like receptor 4-mediated inflammatory signaling that mimics bacterial sepsis. Journal of Korean medical science, 2020. 35(38).

Reviewer 2 Report

I have few concerns to be addressed which are of minor.

What do the authors means by pattern recognition receptors that are being used by the virus?

How do the authors are confident that targeting TLRs, CLRs and other receptors would represent a therapeutic strategy?

 I would find some redundant points throughout the manuscript which may be avoided.

Author Response

What do the authors means by pattern recognition receptors that are being used by the virus?

Pattern recognition receptors are a family of the innate immune receptors, of which TLRs and CLRs are subdivision.  The spike protein of SARS-CoV-2 may be able to bind to some TLRs and CLRs.

How do the authors are confident that targeting TLRs, CLRs and other receptors would represent a therapeutic strategy?

Thank you for pointing this out, as we didn’t give a response as to if we think these could work in the conclusion. As we have mentioned previous therapeutic pharmaceuticals that could be used to treat COVID-19 within the literature we have added the following sentence

Line 947-949: As there is yet to be a vaccine or drug approved to prevent or treat COVID-19, we are confident that focusing on these receptors may be the key dampening severe adverse immunological reactions observed in patients and improving patient outcomes. 

I would find some redundant points throughout the manuscript which may be avoided.

Thank you for this insight. The addition of new information to paragraphs and editing the manuscript again we feel confident that we have omitted redundancy that was present in our first draft.